# Diverse motif ensembles specify non-redundant DNA binding activities of AP-1 family members in macrophages

Gregory J. Fonseca[1], Jenhan Tao[1], Emma M. Westin[1], Sascha H. Duttke[2], Nathanael J. Spann[1], Tobias Strid[1], Zeyang Shen[3], Joshua D. Stender[1], Mashito Sakai[1], Verena M. Link[4], Christopher Benner[2] & Christopher K. Glass[1,2]

Mechanisms by which members of the AP-1 family of transcription factors play non-redundant biological roles despite recognizing the same DNA sequence remain poorly understood. To address this question, here we investigate the molecular functions and genome-wide DNA binding patterns of AP-1 family members in primary and immortalized mouse macrophages. ChIP-sequencing shows overlapping and distinct binding profiles for each factor that were remodeled following TLR4 ligation. Development of a machine learning approach that jointly weighs hundreds of DNA recognition elements yields dozens of motifs predicted to drive factor-specific binding profiles. Machine learning-based predictions are confirmed by analysis of the effects of mutations in genetically diverse mice and by loss of function experiments. These findings provide evidence that non-redundant genomic locations of different AP-1 family members in macrophages largely result from collaborative interactions with diverse, locus-specific ensembles of transcription factors and suggest a general mechanism for encoding functional specificities of their common recognition motif.

[1] Department of Cellular and Molecular Medicine, School of Medicine, University of California San Diego, La Jolla, CA 92037, USA. [2] Department of Medicine, School of Medicine, University of California San Diego, La Jolla, CA 92037, USA. [3] Department of Bioengineering, Jacobs School of Engineering, University of California San Diego, La Jolla, CA 92037, USA. [4] Faculty of Biology, Division of Evolutionary Biology, Ludwig-Maximilian University of Munich, Munich 80539, Germany. These authors contributed equally: Gregory J. Fonseca, Jenhan Tao. Correspondence and requests for materials should be addressed to C.K.G. (email: ckg@ucsd.edu)

G ene expression is controlled by sequence-specific tran-
scription factors (TFs) which bind to promoters and distal
enhancer elements[1–3]. Genome wide studies of regulatory
regions in diverse cell types suggest the existence of hundreds to
thousands of enhancer sites within mammalian genomes. Each
cell type selects a unique combination of ~20,000 such sites that
play essential roles in determining that cell's identity and func-
tional potential[4–7]. Selection and activation of cell-specific
enhancers and promoters are achieved through combinatorial
actions of the available sequence-specific TFs[8–14].

TFs are organized into families according to conserved protein
domains including their DNA binding domains (DBD)[15]. Each
family may contain dozens of members which bind to similar or
identical DNA sequences[16,17]. An example is provided by the AP-1
family, which is composed of 15 monomers subdivided into five
subfamilies based on amino acid sequence similarity: Jun (Jun,
JunB, JunD), Fos (Fos, FosL1, FosL2, FosB), BATF (BATF, BATF2,
BATF3), ATF (ATF2, ATF3, ATF4, ATF7), and Jdp2[18–22]. AP-1
binds DNA as an obligate dimer through a conserved bZIP
domain. All possible dimer combinations can form with the
exception of dimers within the Fos subfamily[23]. The DBD of each
monomer of the AP-1 dimer recognizes half of a palindromic
DNA motif separated by one or two bases (TCASTGA and
TCASSTGA)[16,17,24–26]. Previous work has shown that dimers
formed from Jun and Fos subfamily members bind the same
motif[16]. Given a conserved DBD, and the ability to form hetero-
dimers, it naturally follows that different AP-1 dimers share reg-
ulatory activities. However, co-expressed family members can play
distinct roles[20,27–30]. For example, Jun and Fos are co-expressed
during hematopoiesis, but knockout of Jun results in an increase in
hematopoiesis whereas knockout of Fos has the opposite
effect[20,28–30]. The basis for non-redundant activities of different
AP-1 dimers and heterodimers remains poorly understood.

Specific AP-1 factors have been shown to form ternary com-
plexes with other TFs such as IRF, NFAT, and Ets proteins,
resulting in binding to composite recognition elements with fixed
spacing[31–33]. However, recent studies examining the effects of
natural genetic variation suggested that perturbations in the DNA
binding of Jun in bone marrow-derived macrophages are asso-
ciated with mutations in the motifs of dozens of TFs that
occurred with variable spacing[34]. These observations raise the
general question of whether local ensembles of TFs could be
determinants of differential binding and function of specific AP-1
family members. To explore this possibility, we examined the
genome-wide functions and DNA binding patterns of co-
expressed AP-1 family members in resting and activated mouse
macrophages. In parallel, we developed a machine learning
model, called a transcription factor binding analysis (TBA), that
integrates the affinities of hundreds of TF motifs and learns to
recognize motifs associated with the binding of each AP-1
monomer genome-wide. By interrogating our model, we identi-
fied DNA binding motifs of candidate collaborating TFs that
influence specific binding patterns for each AP-1 monomer that
could not be identified with conventional motif analysis. We
confirmed these predictions functionally by leveraging the natural
genetic variation between C57BL/6J and BALB/cJ mice, and
observing the effects of single nucleotide polymorphisms (SNPs)
and short insertions or deletions (InDels) on AP-1 binding.
Finally, we confirm the models prediction of PPARγ binding
being specifically associated with the selection of a single family
member, Jun, using PPARγ-deficient macrophages.

## Results
**AP-1 members have distinct functions in macrophages**. AP-1
family members are ubiquitously expressed with each cell type

selecting a subset of family members (monomers), which make
up the AP-1 dimer. Each family member shares a conserved DNA
binding and dimerization domain but are dissimilar outside of the
basic leucine zipper (bZIP domain, Fig. 1a). For this study, we will
focus on thioglycollate elicited macrophages (TGEMs). TGEMs,
which are a classical primary macrophage population, are pro-
duced by injection of thioglycolate into the peritoneal space.
Macrophages are then recruited to the peritoneum and can be
easily isolated by flushing the peritoneal cavity 3 days after
treatment. RNA-seq performed on TGEMs revealed ATF3, Jun,
and JunD as the most expressed AP-1 family members under
basal conditions (Veh), (Fig. 1a, Supplementary Fig. 1A). Fol-
lowing activation of TGEMs with Kdo2 lipid A (KLA), a specific
agonist of TLR4[35], there is a marked increase in Fos, Jun, and
JunB expression, consistent with AP-1 family members having
context-specific roles (Fig. 1a).

To examine the regulatory function of individual family
members, knockout cell lines for ATF3, Jun, and JunD were
produced using CRISPR/Cas9-mediated mutagenesis in immor-
talized bone marrow-derived macrophages (iBMDMs). Knockout
efficiency was confirmed by western blotting (Supplementary
Fig. 1B). RNA-seq analysis identified 2496 genes differentially
expressed when comparing the knockout to control cells (false
discovery rate (FDR) < 0.05, fold change > 2, Reads Per Kilobase
of transcript per Million mapped reads (RPKM) ≥ 16; Fig. 1b,
Supplementary Fig. 1C). Clustering of differentially expressed
genes revealed distinct clusters that were affected in individual
knockout cell lines, demonstrating that each family member can
have distinct as well as redundant activity within a single cell type
and corroborating previous studies[20,27–30]. The Jun knockout had
a more modest effect on gene expression than the ATF3 and JunD
knockout (125, 651, and 1564 differentially expressed genes
respectively), suggesting that Jun may have more redundant
activity (Fig. 1b and Supplementary Fig. 1C). Each of the gene
clusters was enriched for Gene Ontology terms for differing
biological functions, including cell cycle, immune effector
process, and NADPH complex assembly (Fig. 1b). Examples of
affected genes are shown in Fig. 1c. *Mmp12* is affected by
knockdown of all three factors, whereas *Marco* and *Fth1* exhibit
minimal changes in expression in ATF3 and Jun KO, but
decreased expression in the JunD KO iBMDMs.

**AP-1 members target distinct in addition to overlapping loci**.
Given the distinct roles of individual family members in reg-
ulating macrophage transcription, we used chromatin immuno-
precipitation followed by deep sequencing (ChIP-seq) to map the
binding of each family member in resting TGEMs treated with
vehicle (Veh) or KLA for 1 h (activated TGEMs). Not surpris-
ingly, these experiments detected a substantial number of binding
sites (n > 10,000, irreproducible discovery rate (IDR) < 0.05) for
family members with the highest mRNA expression (Supple-
mentary Fig. 1A, Supplementary Fig. 2A). ATF3, Jun, and JunD
binding sites were detected in both Veh and KLA treatment
whereas Fos, Fosl2, and JunB bind predominantly after KLA
treatment (Supplementary Fig. 2A). Despite high RNA expression
in Veh treatment, JunB protein expression was not detected in the
nucleus by western blot, explaining a lack of ChIP-seq signal
(Supplementary Fig. 2B). Though ATF4 is highly expressed by
RNA, we were unable to detect ATF4 by ChIP-seq using several
conditions and several different antibodies. Hierarchical cluster-
ing of all 50,664 AP-1 binding sites (Fig. 2a) found in either Veh
or KLA treated TGEMs according to the relative binding strength
of the family members (normalized to a maximum of 1 at each
locus) yielded distinct subclusters that highlight the specific
binding patterns of AP-1 family members as well as the

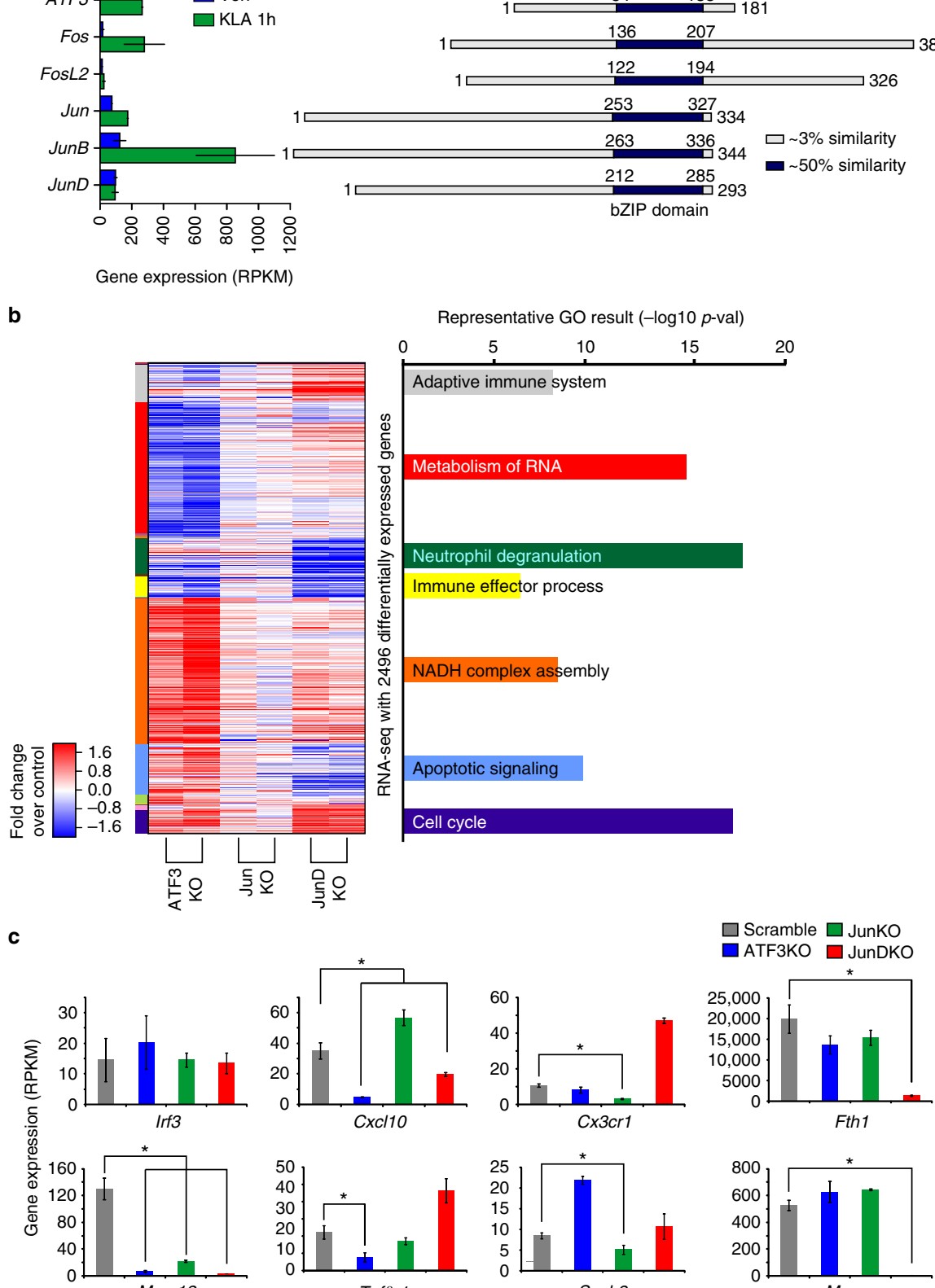

**Fig. 1** AP-1 proteins have overlapping and distinct transcriptional functions in macrophages. **a** Protein alignment of monomers (right) and mRNA expression of monomers in thioglycollate elicited macrophages before and after 1-h Kdo2 lipid A treatment (left). **b** Hierarchical clustering of genes that are differentially expressed in immortalized bone marrow-derived macrophages subjected to CRISPR-mediated knockdown of the indicated AP-1 monomer with respect to scramble control. Expression values are given as the fold change with respect to scramble; values are Z-score normalized across each row. Representative functional annotations for each gene cluster are calculated using Metascape and the enrichment of each term is quantified as the negative log transform of the p-value. **c** Expression of a subset of genes in AP-1 protein knockouts. n=2. End points of the error bars indicate the value from each replicate. * indicates False discovery rate (FDR) < 0.05

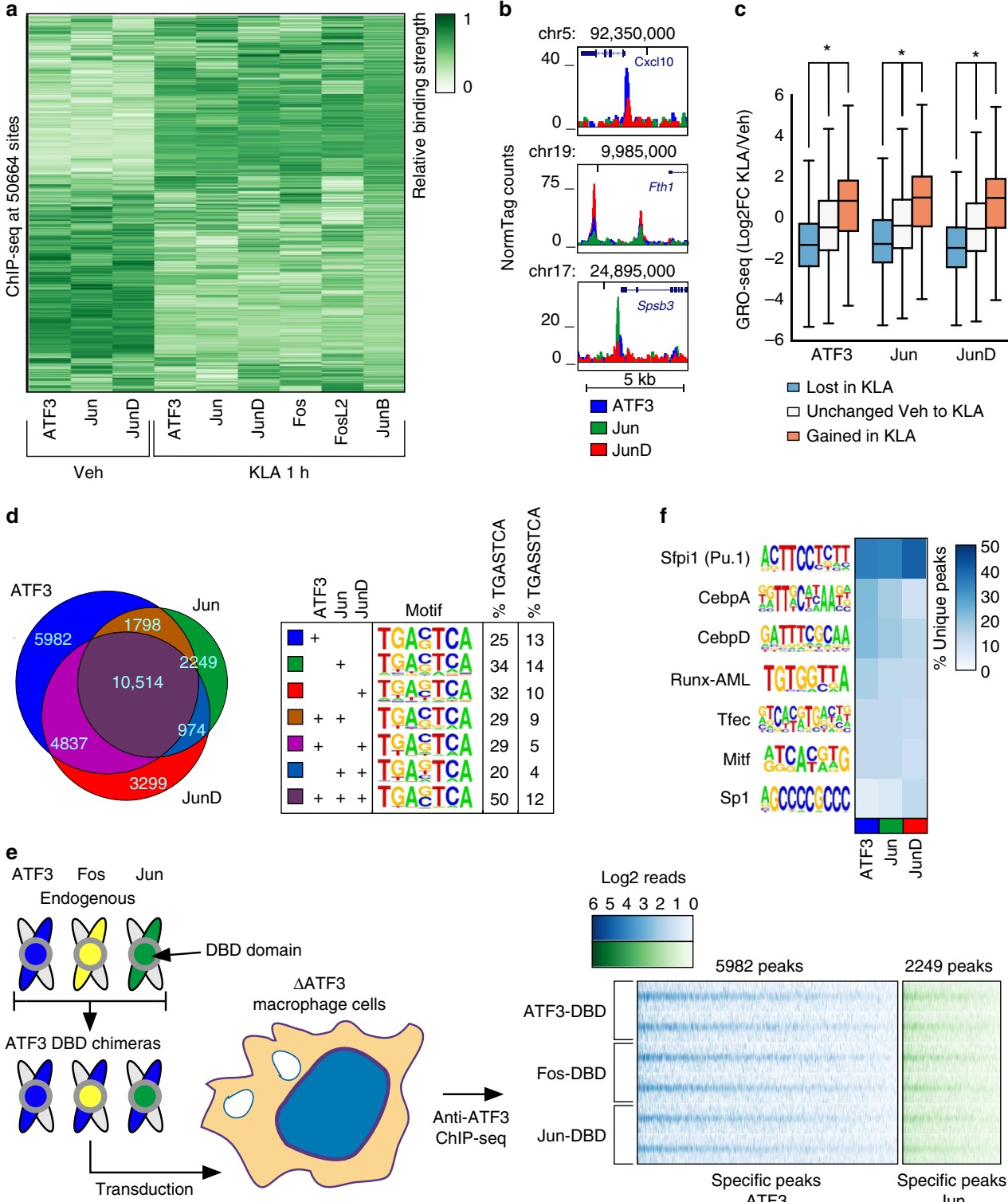

**Fig. 2** AP-1 monomers bind at unique loci that cannot be explained by differences in the DNA binding domain. **a** Hierarchical clustering of the relative strength of binding of each monomer at all AP-1 binding sites in Vehicle and 1-h Kdo2 lipid A (KLA) treatment conditions. **b** Representative browser shots of ChIP-seq peaks for Veh-specific monomers ATF3, Jun, and JunD. **c** Genome run-on sequencing at sites where ATF3, Jun, and JunD were lost, gained, or unchanged after 1 h KLA treatment. **d** Venn diagram of ATF3, Jun, and JunD peaks in Vehicle (left) and table indicating the de novo AP-1 motifs found in each subset of peaks and the percent of peaks in each subset that contain one of the two AP-1 motif variants (right). **e** Binding strength comparison of ATF3 chimeras. The ATF3 DNA binding domain (blue) is replaced by the DNA binding domains of Fos (yellow) or Jun (green) and then transduced into ATF3-deficient immortalized bone marrow-derived macrophage cells with a lentivirus vector (left). The binding of each chimera is shown as a heatmap of ChIP-seq tags centered on ATF3 chimera binding sites (replicates indicated in separate rows) that were found to be specific for ATF3 (blue) or Jun binding in thioglycollate elicited macrophages (green). **f** Heatmap showing the percent of unique binding sites for each monomer that contain a de novo motif calculated from each set of unique peaks. * indicates *p* < 0.01, for all comparisons between Lost in KLA, Unchanged Veh to KLA, and Gained in KLA for each AP-1 family member, independent T-test

reorganization of AP-1 cistromes in KLA treated macrophages (Fig. 2a). Representative regions that show distinct binding patterns of AP-1 family members are shown (Fig. 2b, Supplementary Fig. 2C).

The gain and loss of binding sites of ATF3, Jun, and JunD after KLA treatment provided an opportunity to correlate changes in their DNA occupancy with local changes in enhancer activity. Changes in the expression of enhancer-associated RNAs (eRNAs) are highly correlated with changes in enhancer function and nearby gene expression[11]. To detect eRNAs, we performed genome run-on sequencing (GRO-seq) in TGEMs, which provides a quantitative measure of nascent RNA[36]. We examined GRO-seq signal at ATF3, Jun, and JunD binding sites exhibiting gain, loss, or no change in binding after KLA treatment. In each case, AP-1 occupancy was associated with greater GRO-seq signal (Fig. 2c). These findings suggest that ATF3, Jun, and JunD primarily function as transcriptional activators.

**Member specific loci are associated with a shared DNA motif.** While 10,514 binding sites of ATF3, Jun, and JunD in the vehicle condition are shared by all three factors, a greater number of binding sites (11,530) are not (Fig. 2d). To ensure that the unique sites were not technical artifacts, we ranked the peaks of each family member according to the number of ChIP-seq tags detected and then calculated the percent of peaks that were unique after filtering away binding sites that fell below a given percentile threshold. We found that unique peaks were present even at higher thresholds, supporting our observation that AP-1 family members can bind to distinct loci (Supplementary Fig. 2D).

Using de novo motif enrichment analysis, we observed that the binding motif for each combination of monomers was nearly identical (Fig. 2d). To investigate whether family members preferred either variant of the AP-1 motif, we calculated the percent of peaks bound by each combination of monomers that had the TRE variant of the AP-1 motif (TGASTCA) and the CRE variant of the motif (TGASSTCA)[16,37]. Consistent with previous studies, we found both variants of the AP-1 motif at regions bound by each combination of monomers, but there was a preference for the TRE motif (Fig. 2d)[16]. These results suggest that differences in the AP-1 DBD cannot explain the majority of family member specific binding.

To test the prediction that differences in the AP-1 DBD do not explain binding patterns, we created ATF3 chimeras by replacing the DBD of ATF3 with that of Fos and Jun (Fig. 2e, Supplementary Fig. 2E). The DBDs of these three factors are highly conserved, with identity at 8 and charge conservation at 3 of 11 amino acids directly involved in DNA interaction (Supplementary Fig. 2E)[24]. We transduced expression vectors for ATF3 chimeras with either an ATF3, Fos, or Jun DBD into ATF3 KO iBMDMs and then measured the genome-wide binding patterns of each chimera by performing ChIP-seq using an antibody specific for ATF3 (Fig. 2e). Globally, we observed that the chimeras had stronger binding at ATF3 specific sites in comparison to Jun-specific sites and that each chimera exhibited similar binding across all loci visualized as normalized tag counts in a heatmap (Fig. 2e). Representative browser shots showing similar binding between chimeras are shown at *Cxcl10* and *Spsb1* which are loci specifically bound by ATF3 and Jun, respectively (Supplementary Fig. 2F).

Given that the family members all recognized a common DNA binding motif, we hypothesized that differential interactions with locally bound factors mediated by non-conserved protein contact surfaces may explain unique monomer binding sites. We calculated de novo motifs enriched at the unique peaks for

ATF3, Jun, and JunD individually, and then calculated the percent of each family members specific binding sites that contained a match to each de novo motif. We identified motifs for key TFs in macrophages[10,34] such as PU.1, CEBP, and Runx (Fig. 2f). Composite motifs for AP-1 and IRF or NFAT occurred at similar frequencies at the unique peaks for each family member (~5% and ~3% of peaks, respectively). However, we found no significant differences in the relative enrichment of motifs associated with ATF3, Jun, and JunD specific peaks that would explain their specific binding profiles (Fig. 2f).

**Machine learning links combinations of motifs to TF binding.** Given the robustness of the family member specific peaks (Supplementary Fig. 2D), we considered additional biological mechanisms that might be leveraged for detection of motifs differentially associated with each family member. Current methods for calculating enriched motifs analyze each motif individually despite data demonstrating that TFs bind cooperatively in groups[1,31]. Additionally, collaborative binding by TFs allows for partners to bind to more degenerate motifs, which are ignored in de novo motif analysis[10]. We incorporated these concepts into a machine learning model that relates the presence of multiple TF motifs, which may be degenerate, to the binding of a TF. Machine learning models are often considered difficult to interpret due to their complexity. In building our model, we emphasized simplicity and as a consequence, interpretability.

Figure 3a summarizes our model, TBA. TBA uses logistic regression to learn to distinguish the binding sites of a TF from a set of GC-matched background loci. For each binding site and background locus, TBA calculates the best match to hundreds of DNA binding motifs, drawn from the JASPAR library, and quantifies the quality of the match as the motif score (aka log-likelihood ratio score). To allow for degenerate motifs, all motif matches scoring over zero are considered. The motif scores are then used to train the TBA model to distinguish TF binding sites from background loci. TBA scores the probability of observing binding at a sequence by computing a weighted sum over all the motif scores for that sequence. By considering all motifs simultaneously, TBA can learn to recognize combinations of motifs that are co-enriched at TF binding sites but that are not individually enriched over genomic background. The weight for each motif is learned by iteratively modifying the weights until the models ability to differentiate binding sites from background loci no longer improves. The final motif weight measures whether the presence of a motif is correlated with TF binding. The significance of a given motif can be assigned by comparing the predictive performance of a trained TBA model and a perturbed model that cannot recognize that one motif with the likelihood ratio test.

Machine learning models, including TBA, can be confounded by collinearity, which in our case corresponds to the presence of motifs that are highly similar or redundant[38]. Collinearity can cause inaccurate weight and significance to be assigned to motifs. To assess the extent of collinearity, we calculated the variance inflation factor (VIF)[38] for the scores of each motif in the JASPAR library at AP-1 binding sites. A VIF above 10 would indicate problematic collinearity and that the scores for a motif are highly correlated with the scores of another motif. We found that a substantial number of motifs were collinear with at least one other motif (VIF > 10) (Fig. 3b, c). To address the presence of redundant motifs we clustered the JASPAR library, identifying groups of motifs that are highly similar (Supplementary Fig. 3, colored clades), and merged these motifs together (Pearson correlation > 0.9, Supplementary Fig. 3, Fig. 3a), resulting in a condensed library of 196 motifs formed from 519 JASPAR motifs.

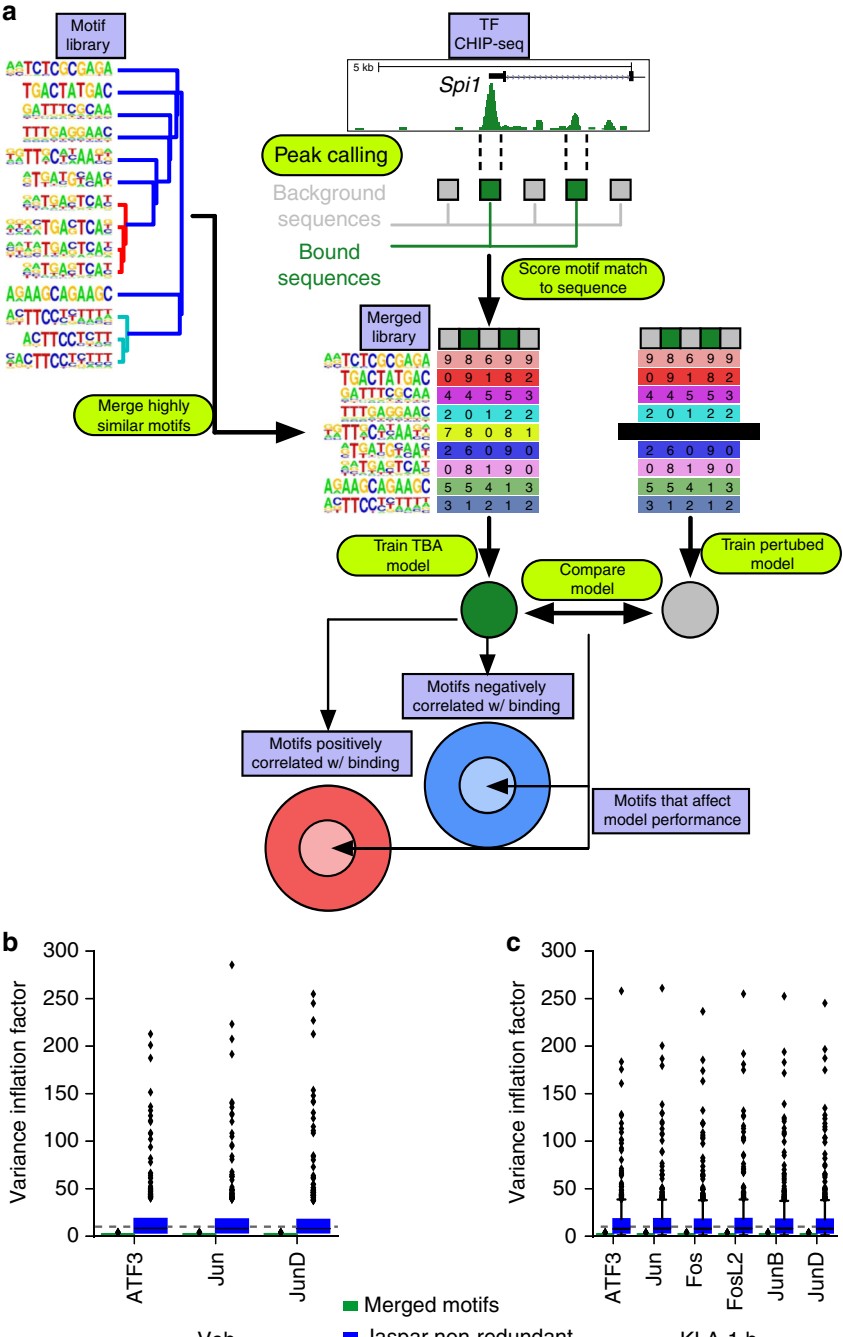

**Fig. 3** TBA, a transcription factor binding analysis. **a** Schematic workflow of TBA. Binding sites for a transcription factor (green boxes) are mixed with random GC-matched background sequences (gray boxes). Motifs from the JASPAR library are merged to create a non-redundant motif library. Motif scores are calculated for all sequences at all binding sites and GC-matched background and then used to train a TBA model. Model weights from the trained model indicate whether a motif is positively or negatively correlated with the occupancy of a transcription factor. The performance of the full model and a perturbed model with one motif removed are compared to identify motifs that are important to the model. The intersection of important motifs that affect model performance and the model weights learned by the classifier can be used to infer the binding partners of a transcription factor. **b**, **c** Distribution of variance inflation factor for each motif in the TBA merged motif library and JASPAR motif library for experiments performed in **b** Vehicle and **c** Kdo2 lipid A treated thioglycollate elicited macrophages

Multiple collinearity was substantially reduced in our condensed library (VIF < 10, Fig. 3b, c).

**TBA identifies combinations of motifs that coordinate AP-1.** To identify motifs associated with specific AP-1 family members, we trained TBA models for each monomer in resting TGEMs,

and probed for differences in the identified motifs. Ranking each motif according to the mean $p$-value, we found that all family members shared a core set of highly significant motifs both positively and negatively correlated with binding (Fig. 4a, i and ii, respectively). The motifs exhibiting strong positive correlation included the AP-1 motif as well as motifs of macrophage collaborative binding partners for AP-1, such as PU.1 and

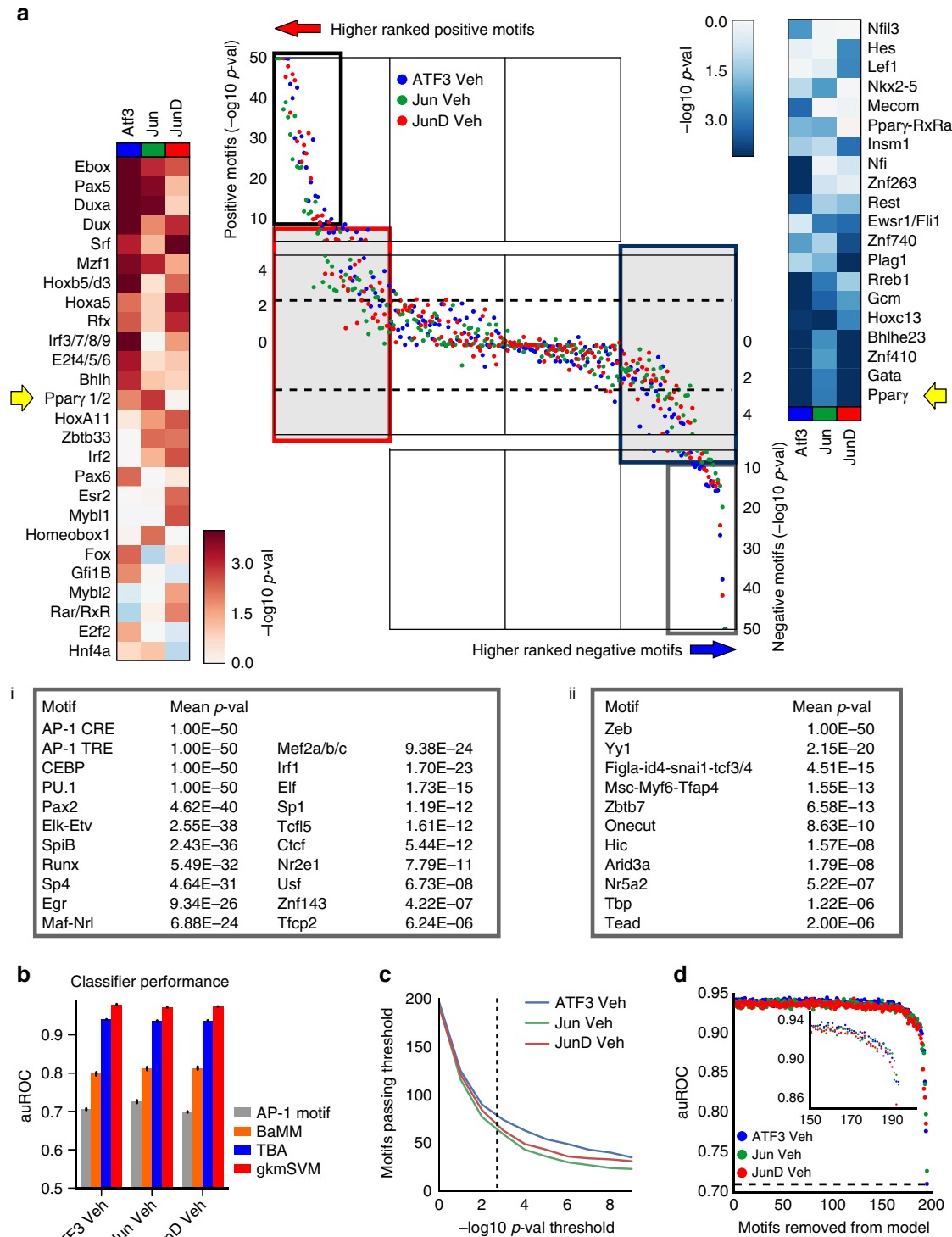

**Fig. 4** Transcription factor binding analysis (TBA) identifies motifs predicted to specify differential AP-1 monomer-binding in resting thioglycollate elicited macrophages. **a** DNA motifs rank order based on the significance of the motif according to the likelihood ratio test. The black box represents the most significant motifs positively correlated with binding for all AP-1 monomers and are listed in (i) and the most significant motifs negatively correlated with binding for all AP-1 monomers are shown in the gray box and are listed in (ii). The significance of motifs positively correlated with binding that show a 100-fold likelihood difference between two monomers are shown on the left heatmap (red); the right heatmap (blue) gives the significance of corresponding motifs negatively correlated with binding. **b** Comparison of the performance of TBA against the AP-1 motif score alone, Bayesian Markov Model (BaMM) motif score, and gapped k-mer SVM as measured by the area under the receiver operating characteristic curve (auROC). Error bars indicate the standard deviation of auROC across 5 cross-validation sets. **c** Number of motifs that pass an in-silico mutagenesis test for significance (the likelihood ratio test comparing the performance of a full model that uses all the motifs and a mutated model with one motif removed) at various p-value thresholds.
**d** Predictive performance of TBA when predicting ATF3, Jun, and JunD binding as motifs are iteratively removed starting from the least important motif based on the weights calculated by TBA. Inset shows performance values beginning at 150 motifs removed where predictive performance begins to drop

CEBP[10,11,34]. To determine a significance threshold for more moderately ranked motifs, we compared significance values calculated by TBA models trained on replicate ChIP-seq experiments. We determined that motifs with a mean $p$-value $< 1e-2.5$ tended to have similar significance values (absolute likelihood ratio ~1, Supplementary Fig. 4A). The motif weights that exceeded this threshold were highly correlated between replicate experiments (Supplementary Fig. 4B). Outside of the core group of motifs shared by all monomers, we observed ~50 motifs with differential affinities (likelihood ratio > 100 between at least 2 monomers) for each monomer as defined by TBA (Fig. 4a, center panel, shaded regions). Differential motifs positively correlated with binding (Fig. 4a, left heatmap in red) included motifs unique to a monomer such as the PPAR half site with Jun. The full PPARγ motif was negatively correlated with both ATF3 and JunD, suggesting that PPARγ positively influences the binding of Jun to a greater extent than the other AP-1 monomers (Fig. 4a, right heatmap in blue). These results suggest that AP-1 monomers have distinct sets of collaborating TFs that affect their binding patterns.

**Evaluation of TF motifs that coordinate AP-1 binding**. To assess whether the additional motifs identified by TBA are useful for identifying AP-1 sites, we compared TBAs ability to predict the binding of each monomer to several other sequence-based approaches. Predicting TF binding using just the AP-1 TRE motif score had the worst performance as measured by the area under the receiver operating characteristic curve (auROC; Fig. 4b). Bayesian Markov Model motifs (BaMM)[39], which assesses dependencies between the positions within the binding motif, improved upon the simple AP-1 motif score by ~15% (Fig. 4b). The TBA model and the gkm-SVM model achieved even higher performance, demonstrating that additional sequences outside of a TFs motif may contribute to binding site selection (Fig. 4b). The performance of gkm-SVM exceeded that of TBA (by ~3%). However, a greater number of motifs related to the binding of a TF can be extracted from TBA in comparison to gkmSVM. The authors of gkm-SVM described a procedure to retrieve up to three PWMs from k-mers ranked by gkmSVM[40], while TBA identified over 50 motifs that passed a significance threshold of $p < 1e-2.5$ (Fig. 4c). To examine the impact of statistically significant ($p < 1e-2.5$) but moderately ranked motifs, we calculated TBAs performance while iteratively removing motifs from the model (starting with the least significant motif) (Fig. 4d). The performance of the model started declining when the motifs from the top 50 were removed, demonstrating that the local sequence environment outside of the AP-1 motif affects AP-1 binding (Fig. 4d, inset).

**Cell type-specific binding preferences of JunD**. To further test the hypothesis that distinct sets of collaborating TFs can affect AP-1 binding, we examined JunD binding in a panel of cell lines. Each cell type expresses a distinct repertoire of TFs that are available as binding partners for JunD. We trained TBA models for ChIP-seq of JunD in each cell line and then extracted the 20 most significant motifs from each model. Motifs which are bound by TFs known to be important for particular cell lines were found to be correlated with JunD binding. For example, the Gata motif was positively correlated with JunD binding in K562 cells, an erythroid lineage erythroleukemia, while Pou motifs (e.g., OCT4) were important in h1-hESCs (Supplementary Fig. 4C)[41]. Differences in the motifs identified by TBA for each cell line corresponded to large differences in the loci bound by JunD (Supplementary Fig. 4D), suggesting that JunD interacts with

different TFs depending on the expressed binding partners available in each cell type[42].

**KLA changes the available TFs that remodel the AP-1 cistrome**. Given that AP-1 binds collaboratively with other TFs, the selection of binding sites for each monomer will depend on the availability of collaborating partners. To study effects of changes in collaborating TF availability, we examined AP-1 binding before and after KLA treatment. Treatment of TGEMs with KLA resulted in 178 mRNAs increasing 2-fold (FDR < 0.05) or greater (Supplementary Fig. 5A). A total of 29 genes encoding TFs with known binding motifs (20 upregulated and 9 downregulated) had a significant change in expression (FDR < 0.05) including AP-1 monomers Fos, Fra2, and JunB (Supplementary Fig. 5A). In addition, TLR4 activation by KLA results in the activation of several latent TFs, including NFκB and interferon regulatory factors (IRFs). Correspondingly, AP-1 monomers showed changes in their global binding patterns with Fos and JunB displaying drastic upregulation in binding sites (Supplementary Fig. 2A, Fig. 5a).

To examine motifs associated with AP-1 binding after KLA treatment, we trained TBA models for each monomer in KLA treated TGEMs. Again, we observed that all AP-1 monomers shared a common group of highly significant motifs positively correlated with binding, including AP-1, CEBP, PU.1, REL, and Egr, and negatively correlated with binding, such as the Zeb1 motif (Supplementary Fig. 5B, Supplementary Table 1, Supplementary Table 2). Many of the moderately ranked motifs showed large differences in significance between the monomers (Supplementary Fig. 5B, Fig. 5c: likelihood ratio > 100).

We found that AP-1 monomers with substantive binding before KLA treatment (ATF3, Jun, and JunD) showed changes in their preference (as measured by the likelihood ratio for each motif when comparing the KLA and Vehicle TBA models) for motifs bound by upregulated TFs such as Rel, Irf3/7/8/9, Irf2, and Nfat (Fig. 5b, likelihood ratio > 10e4). Conversely, downregulated TFs were found to have reduced significance for all AP-1 monomers after 1-h KLA treatment including Usf (Fig. 5b, likelihood ratio <1e−4). AP-1 monomers activated after 1-h KLA treatment (Fos, FosL2, and JunB) (Figs. 2a and 5a) also showed an affinity for the Rel, Nfat, Irf3/7/8/9, and NFκB motifs (Fig. 5b).

To assess the extent to which individual TF motifs could explain the change in binding after KLA treatment, we calculated the correlation of each motifs score to the change in binding after KLA treatment at all loci (Fig. 5c). We found that motifs with large changes in significance when comparing the Vehicle and KLA TBA models for each monomer showed higher correlations to the change in binding after KLA treatment and that these motifs corresponded to well-established TLR4 activated TFs such as Rel, NFAT, and NFκB (Fig. 5b, c)[11,31]. To demonstrate that combinations of TFs can better explain the change in AP-1 binding after KLA treatment, we used TBA to predict the change in binding after KLA treatment. We calculated a predicted change in binding by taking the difference of the predicted binding strength given by the Vehicle and KLA model for each monomer (Fig. 5d–f). We found that TBA could predict the change in binding after KLA treatment better than any individual motif (Fig. 5c).

**Systematic validation of TBA using natural genetic variation**. To validate the results of our machine learning model genome wide, we used natural genetic variation found between C57BL6/J and BALBc/J mice, which differ genetically by ~5 million SNPs and insertions/deletions (InDels)[43]. We have previously shown that mutations which occur within DNA binding motifs can be

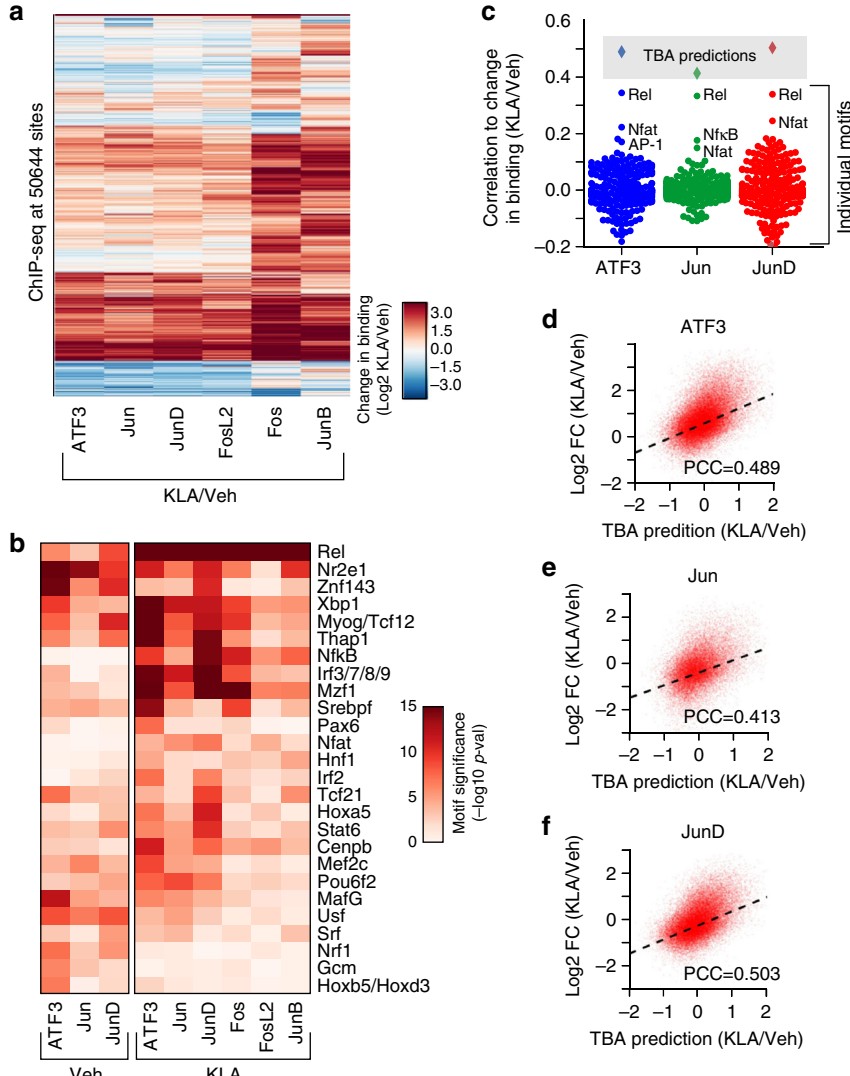

**Fig. 5** AP-1 binding is context-dependent and affected by the availability of binding partners. **a** Heatmap of the change in binding of AP-1 monomers after 1-h Kdo2 lipid A (KLA) treatment quantified as the Log2 ratio of KLA binding to Vehicle binding. **b** Heatmap showing the transcription factor binding analysis (TBA) assigned significance of DNA motifs that had a 10e4 absolute likelihood ratio between the KLA and Vehicle value for each monomer. **c** Pearson correlation of individual motif scores and TBA predictions with the change in binding after 1 h KLA treatment. **d–f** TBA predicted change in ATF3, Jun, and JunD binding after KLA-1 h treatment versus actual change in binding. PCC indicates the Pearson correlation coefficient of TBA predictions to the log2 fold change in binding of each monomer after 1 h KLA treatment

used to predict genetic interactions between TFs[10,34]. We performed ChIP-seq targeting expressed AP-1 monomers, ATF3, Fos, FosL2, Jun, JunB, and JunD in TGEMs isolated from BALB/cJ mice. Mutations can be found in ~17% of each monomers binding sites, and one-third of those loci show strain-specific binding (fold change > 2), as shown for ATF3 (Fig. 6a). These binding differences cannot be attributed to differences in mRNA or protein expression levels, which are highly similar (Supplementary Fig. 6A, B). We observed that TBA models trained on either strain could be used to predict binding in the other with no loss of predictive ability (Supplementary Fig. 6C), suggesting that each monomer, which has identical protein sequence in both strains, interacts with the same repertoire of collaborating TFs in both strains.

To assess the extent to which SNPs/InDels in individual motifs explain strain-specific binding, we calculated the difference between the best matching motif score at every loci between the strains and then calculated the Pearson correlation to the change in binding (Fig. 6b, Supplementary Fig. 6D). Mutations in

individual motifs showed a weak correlation to strain-specific binding (Fig. 6b, Supplementary Fig. 6D). We found that motifs identified with TBA ($p < 1e-2.5$) are enriched at strain-specific peaks in comparison to non-strain-specific peaks, but that mutations in any individual motif do not occur frequently enough to explain the majority of strain-specific binding (Fig. 6c, Supplementary Fig. 6E). We integrated the contributions of multiple motifs to strain-specific binding, by weighting the motif score difference with the TBA calculated weight, and were able to predict strain-specific binding with a 2-fold improvement in performance in comparison to using the AP-1 motif score (Fig. 6b, Supplementary Fig. 6D)

Next, we created a variant of our model, which we call TBA-2Strain, that directly learns from genetic variation (Fig. 6d). TBA-2Strain takes genetic variation as input (quantified as the change in motif scores between the two strains) and the extent of strain-specific binding for each AP-1 monomer. Using TBA-2strain, we predicted strain-specific binding at all binding sites with a mutation (Fig. 6b). In comparison to TBA, TBA-2Strain has

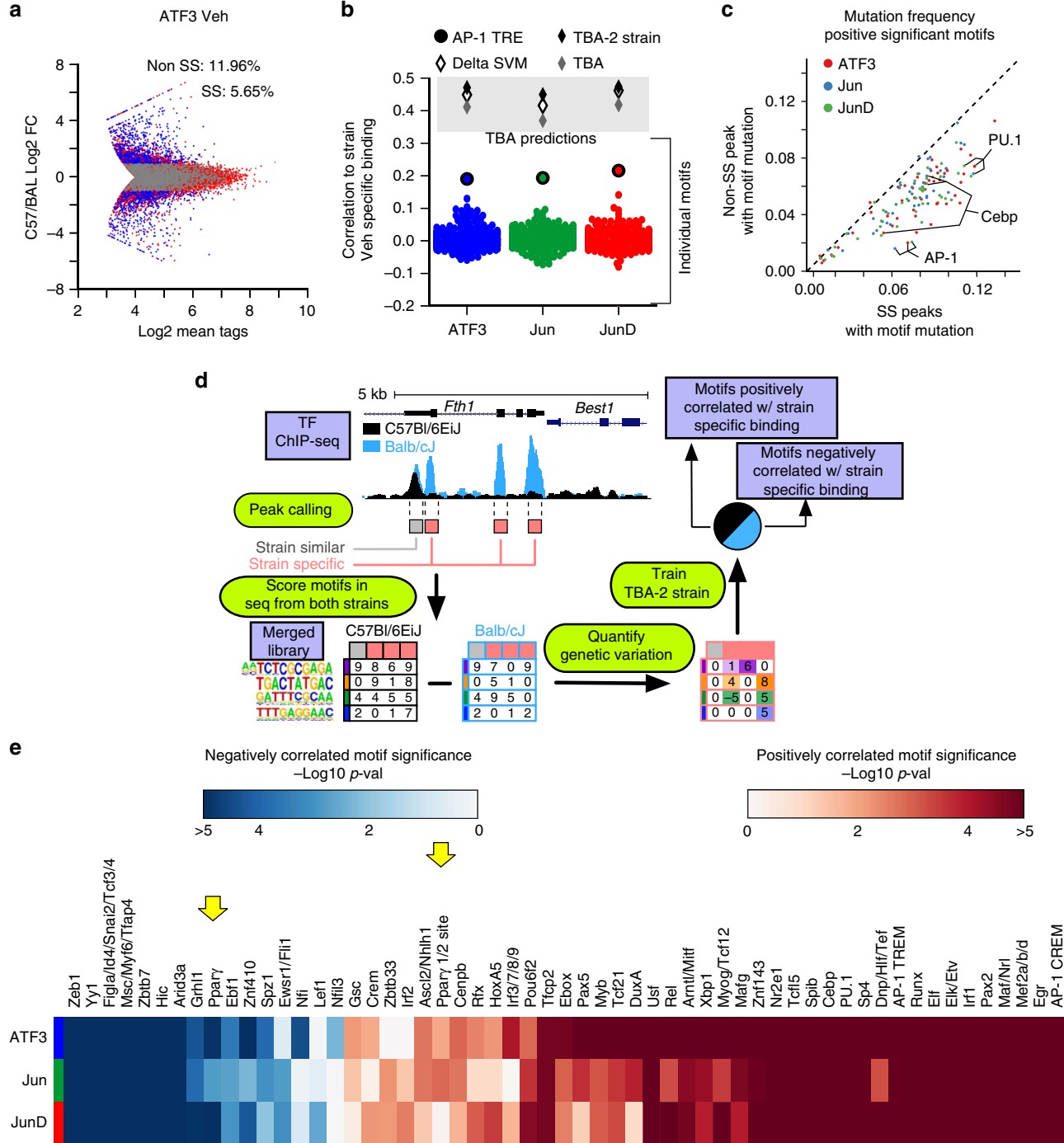

**Fig. 6** Leveraging the effects of genetic variation to validate transcription factor binding analysis (TBA) predictions in resting macrophages. **a** Comparison of the mean strength of binding (number of quantile normalized ChIP-seq tags) for ATF3 in resting thioglycollate elicited macrophages (TGEMs) isolated from C57Bl/6J and Balb/cJ versus the extent of strain-specific binding. Loci with a mutation are indicated in blue (fold change ≥2) when there is strain-specific binding and gray otherwise. **b** Comparison of different models for predicting strain-specific binding of each monomer as measured by the Pearson correlation of a models predictions versus the extent of strain-specific binding in resting TGEMs. Models that integrate multiple motifs deltaSVM, TBA, TBA-2Strain, are represented as diamonds. Individual motifs are indicated using round points. **c** Frequency of mutations in significant motifs (from TBA model, p < 1e−2.5) at strain-specific (fold change ≥2) versus non-strain-specific peaks resting TGEMs. **d** Schematic of TBA-2Strain model. Binding sites for a transcription factor with at least one single nucleotide polymorphism or indel (red boxes) and binding sites with no mutation (gray) are identified. Next, genetic variation is quantified as the difference in the motif scores between the sequences from the two strains and then used as input to train the TBA-2Strain model to predict the extent of strain-specific binding. Model weights from the trained model indicate whether a mutation in a motif is correlated with strain-specific binding. **e** Heatmap of significance values for motifs that intersected between the TBA and TBA-2Strain model for each monomer in resting TGEMs. Blue indicates motifs negatively correlated with binding and red indicates positively correlated motifs

better predictive performance (Fig. 6b). This may be attributed to TBA-2Strain being able to observe sites that contain mutations but do not exhibit strain-specific binding. The ability of TBA-2Strain to predict strain-specific binding improves upon deltaSVM, a state of the art tool for predicting the effect of genetic variation[40] (Fig. 6b, Supplementary Fig. 6D).

We then extracted significant motifs from TBA-2Strain using the F-test ($p < 0.05$) and intersected these motifs with motifs identified by TBA model (Figs. 4a and 6e). We found that the motifs from both models overlapped substantially (Fig. 6e, $p < 0.05$, Fisher's exact test), reinforcing the notion that dozens of motifs contribute to coordinating the targeting of AP-1 monomers. Significance values for motifs identified by both models are shown from resting and activated TGEMs (Fig. 6e, Supplementary Fig. 6F). Notably, the PPARγ half-site was detected by both the TBA and TBA-2Strain models.

**Validation of PPARγ as a preferential modifier of Jun.** TBA and TBA-2Strain predicted that PPARγ is a preferential collaborating TF specific to Jun in resting macrophages (Figs. 4a and 6e). To confirm this prediction, we performed ChIP-seq for ATF3, Jun, JunD, and PPARγ in wild type and PPARγ knockout mouse TGEMs (Fig. 7a–c)[44]. Representative browser tracks are shown for Jun binding in wild-type and PPARγ knockout macrophages (Fig. 7d). The protein expression of ATF3, Jun, and JunD are unchanged in PPARγ knockout TGEMs in comparison to wild type (Fig. 7e). ChIP-seq experiments in PPARγ knockout TGEMs show a marked reduction in Jun binding (Fig. 7a). In contrast, ATF3 and JunD show little change in binding (Fig. 7b, c). We found that PPARγ bound loci where Jun binding is lost in the PPARγ knockout tended to score higher for the PPARγ half site motif in comparison to Jun bound loci that did not overlap with PPARγ binding (independent T-test $p < 5e-05$). To verify the specificity of the Jun antibody we also performed ChIP-seq on Jun in CRISPR mediated Jun knockout iBMDM cells and iBMDM transduced with scramble control. We observed substantial loss of Jun binding in the Jun KO cells in comparison to iBMDM cells transduced with scramble control (12 versus 25,041 peaks detected with IDR < 0.05) (Supplementary Fig. 7). Collectively, these results confirm that PPARγ specifically affects Jun recruitment.

We then probed the interactions between PPARγ and AP-1 family members by co-immunoprecipitation. ATF3, Jun, and JunD co-precipitated with PPARγ (Fig. 7e). As AP-1 binds as a dimer, ATF3 and JunD may be interacting with PPARγ indirectly by dimerizing with Jun. To confirm that Jun is required for interaction of ATF3 and JunD with PPARγ, we performed Co-IP from iBMDM cells in which Jun was knocked out using CRISPR/Cas9 (Supplementary Fig. 1B). We found a loss of interaction between PPARγ and ATF3 or JunD in JunKO cells as compared to scramble control (Fig. 7f). This suggests that ATF3 and JunD do not interact with PPARγ in the absence of Jun.

## Discussion
We demonstrate that AP-1 monomers have both distinct and overlapping transcriptional functions and genome-wide binding patterns in macrophages. Monomer-specific differences in DNA binding are not due to differences in the DBD contact residues as demonstrated by ATF3 chimeras with Jun or Fos DBDs. These observations led us to hypothesize that monomer-specific DNA binding patterns result from locus-specific interactions with different ensembles of collaborating TFs. To address this question, we developed a machine learning model that identified combinations of motifs that are correlated with the binding of a TF. Through this approach, we inferred TF cooperation via the

presence of DNA motifs correlated with the binding of each AP-1 monomer. Leveraging the natural genetic variation found between C57BL/6J and BALB/cJ, we confirmed that mutations in motifs predicted by TBA affect AP-1 binding. Finally, we confirmed that PPARγ plays a preferential role in coordinating Jun binding in TGEMs.

In designing our machine learning model, we optimized for interpretability. We leveraged logistic regression, a relatively simple method, to accurately predict TF binding, and we were able to extract TF motifs underlying these predictions, allowing for the generation of biological hypotheses that can be experimentally validated. A secondary benefit of this approach is that the software can be readily used without specialized computing equipment or a high level of computational understanding. To improve the ability of TBA to robustly identify motifs of interest, we programmatically curated a library that "captures" the core of each motif, thereby mitigating collinearity, which can cause machine learning models to produce inaccurate results. By jointly weighing this library of motifs, TBA enables the detection of combinations of TF binding sites that can predict the distinct and overlapping DNA binding of families of TFs that recognize similar sequences. More broadly, TBA can be applied to predict the effects of mutations on TF binding, and identify determinants of enhancer activation and open chromatin.

There are additional complexities in TF binding and enhancer activation we have not explored. Transcriptional regulation may be encoded by the spacing between motifs as well as the specific arrangement of motifs. Recent neural network architectures, such as CapsuleNets, could allow modeling of these complex properties[45–47]. Although more complex machine learning techniques can be applied to predict TF binding and chromatin state[48–50], it is challenging to extract insights from these models. Efforts to build more advanced methods to extract information from machine learning models will allow not only for interpretation of future models of greater complexity, but also better understanding of existing models[51]. For example, the procedure used by Ghandi et al. to retrieve motifs from gkm-SVM can likely be improved to retrieve additional PWMs[40].

Collectively, our findings suggest two classes of collaborative TFs: (1) highly ranked TFs that are strongly correlated with the binding of all AP-1 monomers, including TFs important to macrophage identity such as such as PU.1 and C/EBPs[10,11,13,52–54] (Fig. 4a, black and gray boxes), and (2) moderately ranked TFs that specify the binding of individual AP-1 monomers (Fig. 4a, red and blue boxes). The former likely consists of TFs that play a role in opening chromatin while the latter class of TFs may allow for tuning the optimal level of transcriptional activation or response. These two classes of motifs were also seen in TLR4 activated macrophages where highly ranked motifs, such as NFκB, were correlated with the binding of all AP-1 family members (Supplementary Table 1), while a large set of moderately ranked motifs distinguished each AP-1 monomer (Supplementary Fig. 5C). Overall, these studies provide evidence that collaborative interactions of TFs allow a single DNA motif to be used in a wide variety of contexts, which may be a general principle for how transcriptional specificity is encoded by the genome.

## Methods
**Statistical analyses**. In Fig. 1c, differences in gene expression was tested using the independent T-test (degree of freedom = 1, two-tailed) on two replicate experiments ($n = 2$). Differentially expressed genes in Fig. 1b were identified using EdgeR[55] with default parameters, and using the cut offs FDR < 0.05 and log2 fold change ≥2. In Fig. 2c, differences between each group (Veh, Shared, and KLA 1 h) were examined using independent T-test (degree of freedom = 1, two-tailed); the number of loci in each group for each monomer are as follows ATF3 (Veh = 1447, Shared = 7460, KLA = 6997), Jun (2390, 3751, 3401), JunD (1351, 5976, 6422). Significance for motifs in Fig. 4a was calculated using the likelihood ratio test

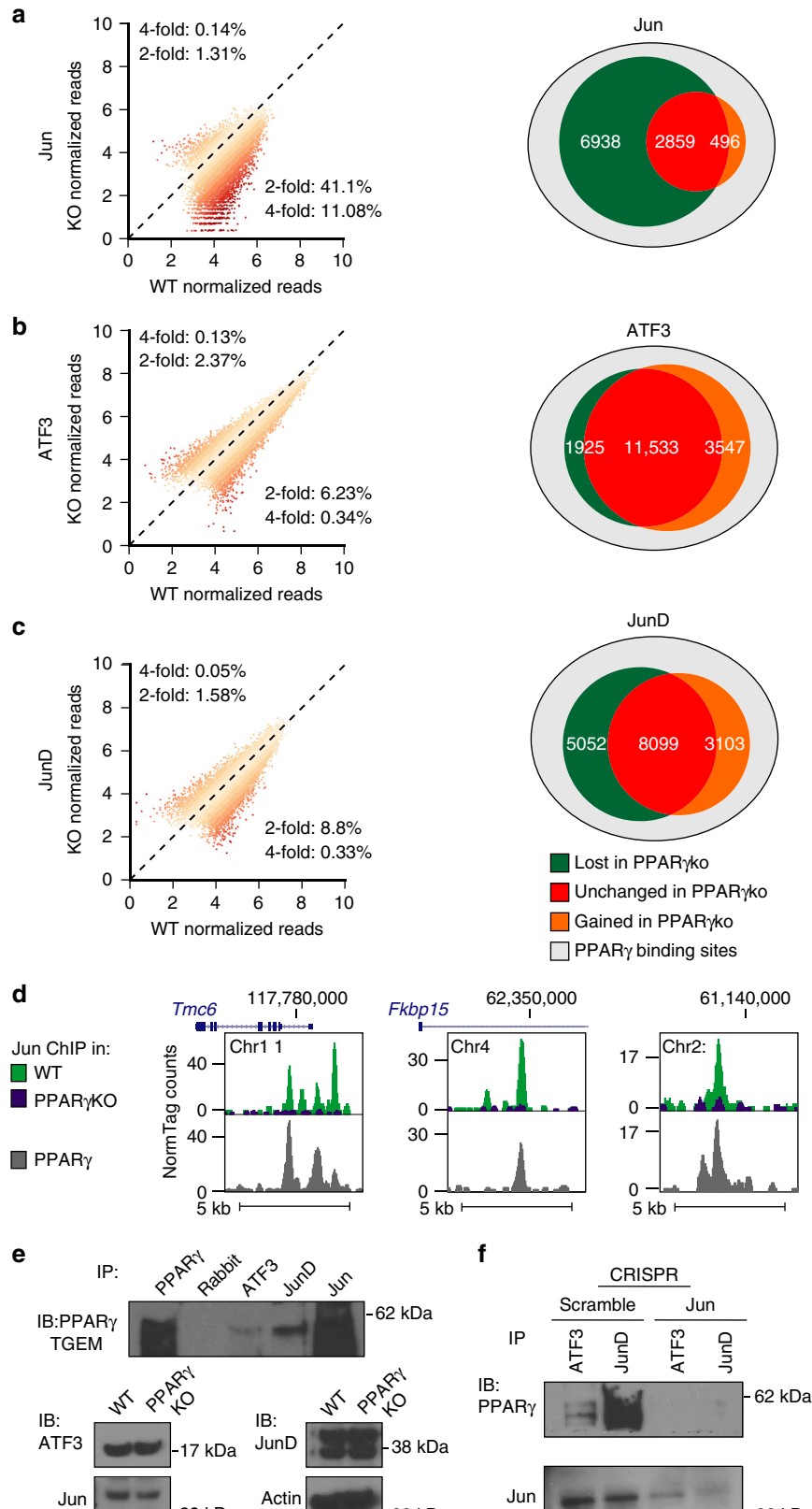

**Fig. 7** The Jun-specific DNA binding program is preferentially altered in PPARγ knockout macrophages. **a–c** Changes in binding strength all binding sites in wild type macrophages in PPARγ-KO macrophages (left) and Venn diagrams summarizing the change in binding at binding sites that overlap with PPARγ (right) for Jun (**a**), ATF3 (**b**), and JunD (**c**). **d** Representative browser shots of Jun in WT and PPARγ-KO thioglycollate elicited macrophages (TGEMs) and PPARγ in WT TGEMs. **e** Western blot analysis of co-immunoprecipitation experiments between AP-1 monomers ATF3, Jun and JunD and PPARγ in TGEMs. **f** Western blot analysis of co-immunoprecipitation experiments between AP-1 monomers ATF3 and JunD and PPARγ in scramble immortalized bone marrow-derived macrophage (iBMDM) or CRISPR-mediated Jun knockout iBMDM

(degree of freedom = 1) comparing the predictions made by the full TBA model and the perturbed TBA model at all loci bound in Veh-treated macrophages for Atf3 ($n = 23,160$), Jun ($n = 15,548$), and JunD ($n = 19,653$). Significance for motifs in Supplementary Fig. 4C was calculated using the likelihood ratio test (degree of freedom = 1) comparing the predictions made by the full TBA model and the perturbed TBA model at all loci bound by JunD in GM12878 ($n = 7451$), H1-hESC ($n = 12,931$), HepG2 ($n = 41,318$), K562 ($n = 47,477$), and SK-N-SH (38,960). Significance for motifs in Fig. 5b, Supplementary Fig. 5B, and Supplementary Fig. 5C was calculated using the likelihood ratio test (degree of freedom = 1) comparing the predictions made by the full TBA model and the perturbed TBA model at all loci bound in KLA treated macrophages for Atf3 ($n = 36,745$), Jun ($n = 17,481$), JunD ($n = 31,641$), Fos ($n = 24,365$), Fosl2 ($n = 10,619$), and JunB ($n = 13,376$). Significance values for Fig. 6f and S6F were calculated using the *F*-test; the number of loci analyzed for monomers in Vehicle-treated macrophages are ATF3 ($n = 4163$), Jun ($n = 3004$), and JunD ($n = 4148$); the number of loci analyzed for monomers in KLA-treated macrophages are: Atf3 ($n = 4577$), Jun ($n = 3232$), JunD ($n = 4366$), Fos ($n = 4477$), and JunB ($n = 3616$).

**Generating custom genome for BALB/cJ**. A custom genome for BALB/cJ by replacing invariant positions of the mm10 genome with alleles reported by the Mouse Genomes Project (version 3 VCF file)[43]. For C57BL/6J the mm10 reference genome from the UCSC genome browser was used. To allow for comparisons between BALB/cJ and C57BL/6J during analysis, the coordinates for the custom genome for BALB/cJ was shifted to match the positions of the mm10 reference genome using MARGE[34]. We did not analyze any reads that fell within deletions in BALB/cJ. Reads that overlapped with an insertion were assigned to the last over-lapping position in the reference strain.

**Analysis of ChIP-seq peaks**. Sequencing reads from ChIP-seq experiments were mapped to the mm10 assembly of the mouse reference genome (or the BALBc/J custom genome) using the latest version of Bowtie2 with default parameters[56]. Mapped ChIP-seq reads to identify putative TF binding sites with HOMER[57] findPeaks command (with parameters -size 200 -L 0 -C 0 -fdr 0.9), using the input ChIP experiment corresponding to the treatment condition. In order to reduce the number of false positive peaks, we calculated the IDR at each peak (using version 2.0.3 of the idr program) with the HOMER peak score calculated for each replicate experiment as the input to IDR and then filtered all peaks that had IDR ≥ 0.05[58]. De novo motifs were calculated with the HOMER findMotifsGenome.pl command with default parameters. Enrichment of de novo motifs was calculated using the findKnownMotifs.pl program in HOMER with default parameters.

Quantification of RNA expression reads generated from RNA-seq experiments were aligned to the mm10 mouse reference genome (or the BALBc/J custom genome) using STAR aligner with default parameters[59]. To quantify the expression level of each gene, we calculated the RPKM with the reads that were within an exon. Un-normalized sequencing reads were used to identify differentially expressed genes with EdgeR[55]; we considered genes with FDR < 0.05 and a change in expression between two experimental conditions two fold or greater differentially expressed. To quantify the expression of nascent RNAs we annotated our ChIP-seq peaks with the number of GRO-seq reads (normalized to 10 million) that were within 500 bps of the peak center using the HOMER annotatePeaks.pl command.

**TBA model training**. For each AP-1 monomer under each treatment condition, we trained a model to distinguish binding sites for each monomer from a set of randomly selected genomic loci. The set of random background loci used to train each model was selected according to the following criteria: (1) the GC content distribution of the background loci matches the GC content of the binding sites for a given monomer, (2) contain no ambiguous or unmappable positions, and (3) the number of background sequences matches the number of binding sites k. For each of the sequences in the combined set of the binding sites and background loci, we calculated the highest log-odds score (also referred to as motif score) for each of the *n* motifs that will be included in the model[60] Motif matches in both orientations were considered. Log-odds scores less than 0 were set to 0. Per standard pre-processing procedures prior to training a linear model, we standardized the log-odds scores for each motif, scaling the set of scores for each motif so that the mean value is 0, and the variance is 1. Standardization scales the scores for all motifs to the same range (longer motifs have a larger maximum score) and also helps to reduce the effect of multi-collinearity on the model training. And so, the features used for training our model is an *n* by 2k matrix of log-odds scores standardized across each row. To generate the corresponding array of labels, we assigned each binding site a label of 1 and each background loci a label of 0. Using this feature matrix, and label array, we trained weights for each motif using an L1 penalized logistic regression model as implemented by the scikit-learn Python package[61]. Motif weights shown in our analysis are the mean values across five rounds of cross-validation, using 80% of the data for training and 20% for testing in each round. Models were trained for ChIP-seqs generated in this study as well as data downloaded from the NCBI Gene Expression Omnibus (accession number GSE46494) and the ENCODE data portal (https://www.encodeproject.org).

**Quantification of multiple collinearity**. To assess the extent of multi-collinearity in the motif score features we used to train our models, we took each feature matrix corresponding to each experiment and calculated the VIF for each motif[38]. To calculate the VIF, we first determine the coefficient of determination, R2, for each motif by regressing the log-odds scores for one motif against the log-odds scores of the remaining motifs. Next using the coefficient of determination, the tolerance for each motif can be calculated as the difference between 1 and the coefficient of determination $(1 - R^2)$. The VIF is the reciprocal of the tolerance $\frac{1}{1 - R^2}$. We used the linear_model module of sklearn Python package to calculate the coefficient of determination.

**Motif clustering and merging**. We scored the similarity of all pairs of DNA sequence motifs by calculating the Pearson correlation of the aligned position probability matrices (PPMs) corresponding to a given pair of motifs[62]. The Pearson correlation for a pair of motifs A and B of length i is calculated using the formula:

$$\frac{\Sigma_i \Sigma_j^4 (A_{ij} - \bar{A}_i)(B_{ij} - \bar{B}_i)}{\sqrt{\left(\Sigma_i \Sigma_j^4 (A_{ij} - \bar{A}_i)\right)^2} \sqrt{\left(\Sigma_i \Sigma_j^4 (B_{ij} - \bar{B}_i)\right)^2}} \quad (1)$$

PPMs were first aligned using the Smith–Waterman alignment algorithm[63]. Shorter motifs are padded with background frequency values prior to alignment. Gaps in the alignment were not allowed and each position in the alignment was scored with the Pearson correlation. The Pearson correlation was then calculated using the optimal alignment. Next, sets of motifs that have PPMs with a Pearson correlation of 0.9 or greater were merged by iteratively aligning each PPM within the set, and then averaging the nucleotide frequencies at each position.

**Assessing significance of motifs for TBA**. p-Values for TBA were calculated using the log-likelihood ratio test. Each motif was removed from the set of features used to train a perturbed TBA model (using five-fold cross-validation). We then used the full model (containing all motifs) and the perturbed model to calculate the likelihood of observing binding on all binding sites and background sequences for a given monomer and all the background regions. The difference in the likelihoods calculated by the full model and the perturbed model was then used to perform the chi-squared test for each motif. The chi-squared test was performed using the scipy python package[64].

**Comparison to other methods**. BaMM motif and gkm-SVM were both run with default parameters. We used the latest version of the larger scale gkm-SVM, LS-GKM (compiled from source code downloaded from https://github.com/Dongwon-Lee/lsgkm on 8/25/16), and BaMM motif; v1.0 downloaded from https://github.com/soedinglab/BaMMmotif[39,65]. Both models were trained using five-fold cross-validation. Model performance was scored using roc_auc_score and precision_score functions from the metrics module of sklearn.

**Predicting changes in AP-1 binding after one-hour KLA treatment**. To predict the change in binding after KLA treatment, we leveraged the motif weights learned for each of the *n* motifs ($w_n$) by a TBA model trained on the Vehicle-treated data ($W_{veh} = [w_{veh,1}...w_{veh,n}]$) and a TBA model trained on the 1-h KLA treated data ($W_{kla} = [w_{kla,1}...w_{kla,n}]$) for each AP-1 monomer. The predicted change in binding for each sequence is then the difference between the dot product of the standardized motif scores calculated for the sequence for each of the *k* binding sites ($S_k = [s_{1,k},...,s_{n,k}]$) with the KLA motif weights and the dot product of the motif scores and the Veh motif weights ($\Delta_{kla-veh,k} = W_{kla} \cdot S_k - W_{veh} \cdot S_k$). Predictions were made for all genomic loci that intersected with a peak for one of the AP-1 monomers in either the vehicle or KLA treatment condition.

**Predicting strain-specific binding with TBA**. To predict strain-specific binding, we leveraged the motif weights learned for each of the *n* motifs ($w_n$) by a TBA model ($W = [w_1,...,w_n]$) for each AP-1 monomer using the C57BL/6J data, and the motif scores calculated for each of the *k* binding sites using the genomic sequence for C57BL/6J and BALBc/J ($S_{C57,k} = [s_{C57,1,k},...,s_{C57,n,k}]$, $S_{BAL,k} = [s_{BAL,1,k},...,s_{BAL,n,k}]$). Next, we computed the difference of the motif scores for C57BL6/J and BALBc/J ($D_n = [s_{C57,n,1} - s_{BAL,n,1}...,s_{C57,n,k} - s_{BAL,n,k}]$) and then standardized the score differences for each motif across all the *k* binding sites that had a mutation when comparing BALBc/J to C57BL/6J, yielding standardized motif score differences for each binding site ($Z_n = \text{standardize}(D_n) = [z_{n,1},...,z_{n,k}]$). Finally, we then made a prediction for strain-specific binding by computing the dot product of the motif weights and the standardized difference of the motif scores between C57BL6/EiJ and BALBc/J for the kth mutated binding site ($\Delta_{C57-BAL} = W \cdot [z_{1,k},...,z_{n,k}]$).

**TBA-2Strain model training**. For each genomic loci that intersected with a peak for one of the AP-1 monomers, in either C57BL/6J or BALBc/J, we calculated the highest log-odds score for each of the *n* motifs that will be included in the model, using the genomic sequence from both strains, yielding a two sets of motif scores for each of the *k* binding sites ($S_{C57,k} = [s_{C57,1,k},...,s_{C57,n,k}]$, $S_{BAL,k} = [s_{BAL,1,k},...,s_{BAL,n,k}]$). Motif matches in both orientations were considered. Log-odds scores less than 0 were set to 0. Using the motif scores, we compute the standardized difference of the motif scores

across the two strains as described in the above section ($Z_n = [z_{n,1}, \dots, z_{n,k}]$). And so, the features used for training our model is an $n$ by $k$ matrix of log-odds scores standardized across each row. Next, we calculated the log2 fold ratio of the number of ChIP-seq reads in C57BL/6J compared to BALBc/J to represent the extent of strain-specific binding. Using this feature matrix, and setting the log2 fold ratio of binding between the two strains as the dependent variable, we trained weights for each motif using linear regression as implemented by the scikit-learn Python package. Motif weights shown in our analysis are the mean values across five rounds of cross-validation, using 80% of the data for training and 20% for testing in each round. Predictions for strain-specific binding can be made using the calculated weights following the procedure in the previous section.

**ChIP protocol.** Protein A and G Dynabeads 50/50 mix from Invitrogen are sued for ChIP (10001D, 10003D). IP mix consists of 20 µL beads/2 µg antibody per 2 million cell ChIP. Antibodies against AP-1 family members were chosen for targeting of non-conserved regions to minimize the potential for non-specific binding. Antibodies are listed in Supplementary Table 3. For preparation, beads were washed with 2× 0.5% BSA–PBS, then beads–antibody were incubated with 0.5% BSA–PBS for at least 1 h on rotator (4 °C). Wash 2× with 0.5% BSA–PBS, then resuspended in dilution buffer (1% Triton, 2 mM EDTA, 150 mM NaCl, 20 mM Tris–HCl (pH 7.4), 1× Protease Inhibitors). Double crosslinking for ChIP: Media was decanted from cells in 10 cm plates, wash once briefly with PBS (RT). Disuccinimidyl glutarate (Pierce Cat # 20593) (diluted in DMSO at 200 mM)/PBS (RT) was used for 10 min. Then formaldehyde was added to a final concentration of 1% for an additional 10 min. Reaction was quenched at 1:10 1 M Tris pH 7.4 on ice. Cells were collected and washed twice with cold PBS, spinning at $1000 \times g$ for 5 min. Nuclei isolation and sonication: Resuspend cell pellets in 1 mL of nuclei isolation buffer (50 mM Tris–pH 8.0, 60 mM KCl, 0.5% NP40) + PI and incubate on ice for 10 min. Centrifuge $2000 \times g$ for 3 min at 4 °C. Resuspend nuclei in 200 µL of fresh lysis buffer (0.5% SDS, 10 mM EDTA, 0.5 mM EGTA, 50 mM Tris–HCl (pH 8)) + PI. Sonication: Nuclei were then sonicated (10 million cells) for 25 min in a Biorupter (settings = 30 s = On, 30 s = Off, Medium) using thin wall tubes (Diagenode Cat# C30010010). After sonication spin max speed for 10 min at 4 °C. ChIP set up: Sonicated DNA was diluted 5× with 800 Dilution Buffer (1% Triton, 2 mM EDTA, 150 mM NaCl, 20 mM Tris–HCl (pH 7.4), 1× protease inhibitors). An aliquot is removed for input samples (5%). Samples ON at 4 °C while rotating. Washing: ChIP are washed 1× with TSE I (20 mM Tris–HCl pH 7.4, 150 mM NaCl, 0.1% SDS, 1% Triton X-100, 2 mM EDTA), 2× with TSE III (10 mM Tris–HCl pH 7.4, 250 mM LiCl, 1% IGEPAL, 1% deoxycholate, 1 mM EDTA), 1× with TE + 0.1% Triton X-100, transfer to new tube and then wash another time with TE + 0.1% Triton X-100. Elution: Elute with 200 µL elution buffer (1% SDS, 10 mM Tris pH 7.5) for 20 min at RT, shaking on the vortex or a nutator or rotator. De-crosslinking: Add 10 µL of 5 M NaCl and incubate ON at 65 °C (or at least 8 h). Clean up samples using Zymo ChIP DNA Clean and Concentrator. Elute in 100 µL. Take 40 µL and proceed to library prep protocol.

**PolyA RNA isolation and fragmentation.** RNA isolation: RNA was isolated using TRIZOL-reagent (Ambion cat# 15596018) and DIRECT-ZOL RNA mini-prep kit (cat# 11-330MB). Poly-A RNA isolation: Use 0.2 total RNA as starting material for ideal mapping efficiency and minimal clonality. Collect 10 µL oligo (dT) (NEB cat# S1419S) beads per RNA sample. Beads were washed twice with 1× DTBB (20 mM Tris–HCl pH 7.5, 1 M LiCl, 2 mM EDTA, 1% LDS, 0.1% Triton X-100). Beads were resuspended in 50 µL of 2× DTBB. 50 µL of beads were mixed with 50 µL RNA and heated to 65 °C for 2 min. RNA-beads were then incubated for 10 min at RT while rotating. RNA-beads were then collected on a magnet and washed 1× each with RNA WB1 (10 mM Tris–HCl pH 7.5, 0.12 M LiCl, 1 mM EDTA, 0.1% LDS, 0.1% Triton X-100) and WB3 (10 mM Tris–HCl pH 7.5, 0.5 M LiCl, 1 mM EDTA). Add 50 µL Tris–HCl pH 7.5 and heat to 80 °C for 2 min to elute. Collect RNA and perform a second Oligo-dT bead collection. After washing the second collection, instead of eluting was 1× with 1× first strand buffer; 250 mM Tris–HCl (pH 8.3), 375 mM KCl, 15 mM MgCl$_2$ (ThermoFisher SSIII kit Cat# 18080093). Fragmentation: Then add 10 µL of 2× first strand buffer plus 10 mM DTT and fragment DNA at 94 °C for 9 min. Collect beads on magnet and transfer eluate containing fragmented mRNA to a new PCR strip. Should recover 10 µL fragmented RNA. First strand synthesis: We mixed fragmented RNA with 0.5 µL random primer (3 µg/µL) Life Tech #48190-011, 0.5 µL oligo-dT (50 µM from SSIII kit), 1 µL dNTPs (10 mM Life Tech, cat 18427088) and 0.5 µL SUPERase-In (ThermoFisher Cat#AM2696) and heat 50 °C for 1 min. Immediately place on ice. We then added 5.8 µL ddH$_2$O, 0.1 µL actinomycin (2 µg/µL Sigma cat#A1410), 1 µL DTT (100 mM Life Tech cat# P2325), 0.2 µL of 1% Tween and 0.5 µL of Superscript III and incubate 25 °C for 10 min, then 50 °C for 50 min. Bead clean up: We added 36 µL of RNAClean XP (Ampure XP) and mixed, incubating for 15 min on ice. The beads were then collected on a magnet and washed 2× with 75% ethanol. Beads were then air-dried for 10 min and elute with 10 µL nuclease-free H$_2$O. Second strand synthesis. 10 µL of cDNA/RNA was mixed with 1.5 µL 10× Blue Buffer (Enzymatics cat# B0110L), 1 µL dUTP/dNTP mix (10 mM Affymetrix cat# 77330), 0.1 µL dUTP (100 mM Affymetrix cat# 77206), 0.2 µL RNase H (5 U/µL Enzymatics cat# Y9220L), 1 µL DNA polymerase I (10 U/µL Enzymatics cat#P7050L), 0.15 µL 1% Tween-20 and 1.05 µL nuclease-free water. Reaction was incubated at 16 °C for 2.5 h. Bead clean up: DNA was purified by adding 1 µL Seradyn 3 EDAC SpeedBeads

(Thermo 6515-2105-050250) per reaction in 28 µL 20% PEG8000/2.5 M NaCl (13% final concentration) and incubating at RT for 10 min. Beads were then collected on a magnet and washed 2× with 80% ethanol. Beads were air-dried for 10 min and eluted in 40 µL of nuclease-free water. DNA is ready for library prep.

**Library prep protocol.** dsDNA end repair: We mixed 40 µL of DNA from ChIP or RNA protocols with 2.9 µL of H$_2$O, 0.5 µL 1% Tween-20, 5 µL 10× T4 ligase buffer (Enzymatics cat# L6030-HC-L), 1 µL dNTP mix (10 mM Affymetrix 77119), 0.3 µL T4 DNA pol (Enzymatics P7080L), 0.3 µL T4 PNK (Enzymatics Y9040L), 0.06 µL Klenow (Enzymatics P7060L) and incubated for 30 min at 20 °C. 1 µL of Seradyn 3 EDAC SpeedBeads (Thermo 6515-2105-050250) in 93 µL 20% PEG8000/2.5 M NaCl (13% final) was added and incubated for 10 min. Bead clean-up: Beads were collected on a magnet and washed 2× with 80% ethanol. Beads were air-dried for 10 min and eluted in 15 µL ddH$_2$O. dA-Tailing. DNA was mixed with 10.8 µL ddH$_2$O, 0.3 µL 1% Tween-20, 3 µL Blue Buffer (Enzymatics cat# B0110L), 0.6 µL dATP (10 mM Tech 10216-018), 0.3 µL Klenow 3-5 Exo (Enzymatics P7010-LC-L) and incubated for 30 min at 37 °C. 55.8 µL 20% PEG8000/2.5 M NaCl (13% final) was added an incubated for 10 min. Then bead clean up was done. Beads were eluted in 14 µL. Y-shape adapter ligation. Sample was mixed with 0.5 µL of a BIOO barcode adapter (BIOO Scientific cat# 514104), 15 µL Rapid Ligation Buffer (Enzymatics cat@ L603-LC-L), 0.33 µL 1% Tween-20 and 0.5 µL T4 DNA ligase HC (Enzymatics L6030-HC-L) and incubated for 15 min at RT. 7 µL of 20% PEG8000/2.5 M NaCl was added and incubated for 10 min at RT. Bead clean up was performed and beads were eluted in 21 µL. 10 µL was then used for PCR amplification (14 cycles) with IGA and IGB primers (AATGATACGGCGA CCACCGA, CAAGCAGAAGACGGCATACGA).

**GRO-seq.** Nascent transcription was captured by global nuclear run-on sequencing (GRO-seq). Nuclei were isolated from TGEMs using hypotonic lysis (10 mM Tris–HCl (pH 7.5), 2 mM MgCl$_2$, 3 mM CaCl$_2$; 0.1% IGEPAL CA-630) and flash frozen in GRO-freezing buffer (50 mM Tris–HCl (pH 7.8), 5 mM MgCL$_2$, 40% Glycerol). Run-on. $3$–$5 \times 10^6$ BMDM nuclei were run-on with BrUTP-labeled NTPs with 3× NRO buffer (15 mM Tris–Cl (pH 8.0), 7.5 mM MgCl$_2$, 1.5 mM DTT, 450 mM KCl, 0.3 U/µL of SUPERase In, 1.5% Sarkosyl, 366 µM ATP, GTP (Roche), Br-UTP (Sigma 40 Aldrich) and 1.2 µM CTP (Roche, to limit run-on length to ~40 nucleotides)). Reactions were stopped after 5 min by addition of 500 µL Trizol LS reagent (Invitrogen), vortexed for 5 min and RNA extracted and precipitated as described by the manufacturer. RNA pellets were resuspended in 18 µL ddH$_2$O + 0.05% Tween (dH$_2$O + T) and 2 µL fragmentation mix (100 mMZnCL$_2$, 10 mM Tris–HCl (pH 7.5)), then incubated at 70 °C for 15 min. Fragmentation was stopped by addition of 2.5 µL of 100 mM EDTA. BrdU enrichment. BrdU enrichment was performed using BrdU antibody (IIB5) and AC beads (Santa Cruz, sc-32323 AC, lot #A0215 and #C1716). Beads were washed once with GRO binding buffer (0.25× saline-sodium-phosphate-EDTA buffer (SSPE), 0.05% (vol/vol) Tween, 37.5 mM NaCl, 1 mM EDTA) + 300 mM NaCl followed by three washes in GRO binding buffer and resuspended as 25% (vol/vol) slurry with 0.1 U/µL SUPERase-in. To fragment RNA, 50 µL cold GRO binding buffer and 40 µL equilibrated BrdU antibody beads were added and samples slowly rotated at 4 °C for 80 min. Beads were subsequently spun down at $1000 \times g$ for 15 s, supernatant removed and the beads transferred to a Millipore Ultrafree MC column (UFC30HVNB; Millipore) in 2 × 200 µL GRO binding buffer. The IP reaction was washed twice with 400 µL GRO binding buffer before RNA was eluted by incubation in 200 µL Trizol LS (ThermoFisher) under gentle agitation for 3 min. The elution was repeated a second time, 120 µL of dH$_2$O + T added to increase the supernatant and extracted as described by the manufacturer. End repair and decapping: For end-repair and decapping, RNA pellets were dissolved in 8 µL TET (10 mM Tris–HCl (pH 7.5), 1 mM EDTA, 0.05% Tween 20) by vigorous vortexing, heated to 70 °C for 2 min and placed on ice. After a quick spin, 22 µL Repair master mix (3 µL 10× PNK buffer, 15.5 µL dH$_2$O + T, 0.5 µL SUPERase-In RNase Inhibitor (10 U), 2 µL PNK (20U), 1 µL RppH (5U)) was added, mixed and incubated at 37 °C for 1 h. To phosphorylate the 5′end, 0.5 µL 100 mM ATP was subsequently added and the reactions were incubated for another 45 min at 37 °C (the high ATP concentration quenches RppH activity). Following end repair, 2.5 µL 50 mM EDTA was added, reactions mixed and then heated to 70 °C for 2 min before being placed on ice. A second BrdU enrichment was performed as detailed above. RNA pellets were dissolved in 2.75 µL TET + 0.25 µL Illumina TruSeq 3′Adapter (10 µM), heated to 70 °C for 2 min and placed on ice. 7 of 3′master mix (4.75 µL 50% PEG8000, 1 µL 10× T4 RNA ligase buffer, 0.25 µL SUPERase-In, 1 µL T4 RNA Ligase 2 truncated (200U; NEB)) was added, mixed well and reactions incubated at 20 °C for 1 h. Reactions were diluted by addition of 10 µL TET + 2 µL 50 mM EDTA, heated to 70 °C for 2 min, placed on ice and a third round of BrUTP enrichment was performed. RNA pellets were transferred to PCR strips during the 75% ethanol wash and dried. Samples were dissolved in 4 µL TET (10 mM Tris–HCl (pH 7.5), 0.1 mM EDTA, 0.05% Tween 20) + 1 µL 10 µM reverse transcription (RT) primer. To anneal the RTprimer, the mixture was incubated at 75 °C for 5 min, 37 °C for 15 min and 25 °C for 10 min. To ligate the 5′ Illumina TruSeq adapter, 10 µL 5′master mix (1.5 µL dH$_2$O + 0.2% Tween 20, 0.25 µL denatured 5′TruSeq adapter (10 µM), 1.5 µL 10× T4 RNA ligase buffer, 0.25 µL SUPERase-In, 0.2 µL 10 mM ATP, 5.8 µL 50% PEG8000, 0.5 µL T4 RNA ligase 1 (5U; NEB)) was added and reactions were incubated at 25 °C for 1 h. Reverse transcription was

performed using Protoscript II (NEB) (4 μL 5× NEB FirstStrand buffer (NEB; E7421AA), 0.25 μL SUPERase-In, 0.75 μL Protoscript II (150U; NEB)) at 50 °C for 1 h. After addition of 30 μL PCR master mix (25 μL 2× LongAmp Taq 2× Master Mix (NEB), 0.2 μL 100 μM forward primer, 2.8 μL 5 M betaine and 2 μL 10 μM individual barcoding primer), mixtures were amplified (95 °C for 3 min, (95 °C for 60 s, 62 °C for 30 s, 72 °C for 15 s) x13, 72 °C for 3 min). PCR reactions were cleaned up using 1.5 volumes of SpeedBeads (GE Healthcare) in 2.5 M NaCl/20% PEG8000. Libraries were size selected on PAGE/TBE gels to 160–225 base pairs. Gel slices were shredded by spinning through a 0.5 mL perforated PCR tube placed on top of a 1.5 mL tube. 150 μL Gel EB (0.1% LDS, 1 M LiCl, 10 mM Tris–HCl (pH 7.8)) was added and the slurry incubate under agitation overnight. To purify the eluted DNA, 700 μL Zymogen ChIP DNA binding buffer was added into the 1.5 mL tube containing the shredded gel slice and the Gel EB, mixed by pipetting and the slurry transferred to a ZymoMiniElute column. Samples were first spun at 1000 × g for 3 min, then 10,000×g for 30 s. Flow through was removed, and samples washed with 200 μl Zymo WashBuffer (with EtOH). Gel remainders were removed by flicking and columns washed by addition of another 200 μL Zymo WashBuffer (with EtOH). Flow through was removed, columns spun dry by centrifugation at 14,000 × g for 1 min and DNA eluted by addition of 20 μL pre-warmed Sequencing TET (10 mM Tris–HCl (pH 8.0), 0.1 mM EDTA, 0.05% Tween 20). Libraries were sequenced.

**Western blotting**. Cells were lysed with Igepal lysis buffer (50 mM Tris pH 8.0, 150 mM NaCl, 0.5% Igepal) and protein concentrations were determined with BioRad protein assay reagent using BSA as a standard. Proteins were separated on NuPage 4–12% Bis–Tris gradient gels (Invitrogen) and transferred onto a nitro-cellulose membrane (Amersham). Membranes were blocked in TBS with 0.1% Tween-20 and 5% BSA. Membranes were blotted with the indicated primary overnight at 4 °C. Horseradish peroxidase-conjugated secondary antibodies were detected using ECL plus western blotting detection system (Amersham).

**Animals and cell culture**. TGEMs were collected 3 days after injection from male 8-week C57Bl/6J, or BALB/cJ mice, and plated at $20 \times 10^6$ cells per 15 cm Petri dish in DMEM plus 10% FBS and 1× penicillin–streptomycin. One day after plating, cells were supplemented with fresh media and treated with PBS (Veh) or 100 ng/mL KLA for 1 h, and then directly used for downstream analyses. iBMDM are produced by infection of BMDM with a retrovirus containing myc and Braf V600E[66]. The immortalized cells are then grown out over several weeks. All animal experiments were performed in compliance with the ethical standards set forth by University of California, San Diego Institutional Annual Care and Use Committee (IUCAC).

**Lentivirus production**. pLentiguide was modified to contain a U6-bsmbi-spgRNA scaffold and a CMV promoter driving tagBFP2. 2 CRISPR guides were inserted for each target via PCR amplification with the H1 promoter (bsmbi site/guide1/scaf-fold/H1 promoter/guide 2/bsmb1 site) for a total of 2 guides per virus (U6 and H1 driven) (Supplementary Table 4). Virus was made with pVSVg/ppAX2 system. Two days post transfection, media was collected and centrifuged at 4 °C for 2 h at 20,000 × g. Cell pellet was reconstituted overnight at 4 °C in OPTI-MEM and stored at −80 °C.

**Production of CRISPR KO iBMDMs**. KO iBMDMs were produced using lentiviral infection. iBMDM-CAS9-IRES-EGFP were infected with MOI 100, as measured on 293T cells, with Lentiblast (OZ biosciences) (5 μL each reagent) in OPTI-MEM. This was then centrifuged at 1300g for 1 h at room temperature. Media was then removed and cells were supplemented in bone marrow media (30% L-cell, 20% FBS, 1% penicillin/streptomycin in DMEM) for 2 days. Cells were then sorted for infection by expression of a transgene on the viral sequence (tagBFP2).

**Reporting summary**. Further information on experimental design is available in the Nature Research Reporting Summary linked to this article.

**Code availability**. All algorithms relating to training and testing our model, TBA, has been implemented using Python. Source code and executable files are available at: https://github.com/jenhantao/tba.

## Data availability

Data generated for this study has been deposited to the NCBI Gene Expression Omnibus (GEO) under the accession number GSE111856. Previously published data was downloaded from GEO (accession number GSE46494) and the ENCODE data portal (https://www.encodeproject.org). A reporting summary for this Article is available as a Supplementary Information file. The individual data points underlying all figures and tables that report average values as well as uncropped versions of gels and blots are available as Source Data file.

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

## Acknowledgements

The authors thank L. Van Ael for assistance with manuscript preparation and J. Collier, M. Pasillas, and Z. Ouyang for technical assistance. These studies were supported by NIH Grants DK091183, CA17390, and GM085764 and Leducq Transatlantic Network Grant 16CVD01 to C.K.G. DNA sequencing was supported by NIH Grant DK063491. S.H.D. is a CRI-Irvington Postdoctoral Fellow. T.S. was supported by the Swedish Society for Medical Research. G.J.F. was supported by a Canadian Institute of Health Research Postdoctoral Fellowship, FME-135475. M.S. was supported by the Manpei Suzuki Diabetes Foundation of Tokyo, Japan, and the Osamu Hayaishi Memorial Scholarship for Study Abroad, Japan.

## Author contributions

G.J.F., J.T., and C.K.G. conceived the study. G.J.F., J.T., E.M.W., S.H.D., J.D.S., T.S., N.J.S., M.S., and V.M.L. performed experiments. J.T., Z.S., G.J.F., C.B., and C.K.G. analyzed data. G.J.F., J.T., and C.K.G. wrote the manuscript with contributions from C.B.

## Additional information

**Competing interests:** The authors declare no competing interests.

