## [Peer Review File · Nature Communications]

Reviewer #1 (Remarks to the Author):

In this study, Fonseca et al. investigated the molecular bases for shared as well as specific ChIP-seq profiles of different AP-1 family members in basal and KLA-activated macrophages. Understanding how transcription factors with identical DNA binding specificities are recruited to different sites is in fact a classical and still unsolved problem that the authors tackled in a lucid and effective manner by combining machine learning approaches with mouse genetics.

The main conclusion of the study is that DNA binding specificities in vivo are the result of the availability of partner transcription factors that selectively promote recruitment of a specific AP-1 family member.

Overall, the approach, the data and the interpretation of the results are convincing and the manuscript represents an important contribution to the field.

I have only a couple of comments for improvement:

1. It is unclear if the KO iBMDMs used represent different clones or polyclonal populations of deleted cells. Moreover, no details are provided regarding the generation of these immortalized cells.
2. One point I do not fully understand is why no motifs for the cooperating transcription factors identified by machine learning were not retrieved in a standard over-representation analysis (page 6, bottom). I assume this is due to the low fraction of specific events that are explained by each cooperating TF, but it would be important to clarify this issue.

Reviewer #2 (Remarks to the Author):

Fonseca et al study the sequence features that are responsible for distinct DNA binding activities of different AP-1 family members in thioglycolate-elicited macrophages before and after stimulation with a TLR4 agonist. They do this by first identifying the binding sites of six AP-1 family members (ATF3, Jun, JunD, Fos, FosL2 and JunB) using ChIP-seq. They find that there are substantial differences in the binding patterns of all of the family members. To understand the reasons for these differences, they develop a simple logistic regression model to identify transcription factor motifs that are predictive for the binding of different AP-1 members. They also perform a number of validation experiments in immortalised bone-marrow derived macrophages confirm their findings. Overall, the paper is well written and provides novel insights into the binding specificities of AP-1 family members as well as confirming well-established models of collaborative interactions between TFs. The paper also highlights that a potentially large number of binding partners can influence the binding of a single TF - an observations that is currently not well appreciated in the field. However, all of these results rely on a crucial assumption that the antibodies used for the ChIP-seq

experiments are highly specific to their target proteins. Moreover, important experimental details to judge the validity of this assumption are currently missing from the manuscript.

Major concerns

The quality of ChIP-seq experiments crucially depend on the quality and specificity of the antibodies used (<https://www.nature.com/news/reproducibility-crisis-blame-it-on-the-antibodies-1.17586>). Non-specific antibodies can lead to false positive results and multiple Nature group journals are now requiring the authors to fill a reporting checklist describing which antibodies were used and how they were validated. The current version of the manuscript does not contain any information about the antibodies used (manufacturer name, lot number, etc) and how their specificity for the different AP-1 family members was validated. Thus, it is not clear if the differences in ChIP-seq signal observed between the six AP-1 family members are due to intrinsic binding properties of the TFs or non-specific binding of the antibodies to other transcription factors. At a minimum, the authors should clearly state which antibodies were used for the study and how their binding specificities were validated (either by the manufacturer or in the lab of the authors). If no evidence for validation can be provided, this should clearly be stated in the manuscript, explicitly acknowledging how this might influence the interpretation of the results.

Similarly, the authors report that the PPAR half-site motif is specifically enriched among the ChIP-seq peaks of the Jun TF. However, the same result would also be consistent with a scenario in which the Jun antibody exhibits low specificity for the PPAR γ factor which could drive the signal. The best validation would be to perform the ChIP-seq experiment for Jun in Jun-deficient cells to demonstrate that without the target protein present the ChIP-seq signal is reduced to background levels. However, if this is not feasible, the authors should seek other means of validation such as using a different antibody for Jun or demonstrating that in PPAR γ knockout cells the Jun signal decreases specifically at sites that are also bound by Jun and not at other PPAR γ binding sites across the genome.

Page 2, pg 3. The authors state that the experiments were performed in thioglycolate-elicited macrophages (TGEMs). However, it is not clear why TGEMs were chosen over other macrophage models. Moreover, it is not clear why the authors did not use BMDMs for their main experiments, given that this cell type was used for all of the knockout experiments. While I agree that this difference in cell types is unlikely to change the results, it would nevertheless be nice to have a justification for the experimental design somewhere in the paper. Secondly, since Nature Communications is a journal of general interest, it would be useful to add 1-2 sentence description of what the TGEMs are and how they are derived. People outside of the mouse macrophage field (but still interested in determinants of TF binding) might not be familiar with all of the different mouse macrophage models available.

Minor issues

Throughout the main text of the paper, it is not clear which machine learning algorithm is being used. It is only in the discussion where the authors mention that they have used logistic regression. Although it is reasonable to leave implementation details for the Methods section, it would be beneficial to introduce the main learning algorithm earlier in the paper. For example, the caption for Figure 3 could contain this information.

Can you reproduce the PPAR γ result from the gkm-SVM model? For example, by looking at the k-mers that have high predictive power for Jun binding? Can you extract k-mer weights from the gkm-SVM model?

p19 line 6 from the bottom: Missing 'to' in: 'To assess the extent multi-collinearity in the motif score features we used to train our models.'

p20, pg 2. Since PPMs are matrices and not vectors, it is not clear how the Pearson correlation between the two motifs is defined. Please clarify. Also, why did you decide to use Pearson correlation as the similarity measure for motif clustering, given that multiple other alternatives are available?

p20, pg 4. Please provide the exact versions of the 'latest' gkm-SVM, LS-GKM and BaMM software packages that were used in this study.

p17, pg 2 from bottom. In the discussion of more complex deep neural network models, you state that they are often difficult to interpret. While this is true in general, considerable progress has been made in this area recently. For example, you might find this recent paper from Peyton Greenside et al relevant (<https://www.biorxiv.org/content/early/2018/04/17/302711>).

Reviewer #3 (Remarks to the Author):

Fonseca et al describe genome-wide transcriptional regulation of AP-1 in thioglycolate-elicited macrophages and immortalized bone marrow derived macrophages. They show that (1) AP-1 family members have distinct regulatory roles and can target monomer-specific loci in addition to overlapping ones (2) monomer-specific binding is not explained by differences in the DNA-binding domain but rather a differential interaction of locally-bound factors described as ensembles of collaborating TFs (PU.1, CEBP, RUNX). They then developed a machine learning approach (TBA – transcription factor Binding Analysis) with reduced multiple collinearity, which shows relatively higher performance when compared with existing sequence-based approaches with the exception of gkm-SVM. Outside the logical flow of the manuscript, the authors went on demonstrating the cell-type specific binding preferences of JunD. Finally, the manuscript shows that (3) TLR4 activation with KLA treatment reshapes the AP-1 cistrome which could be predicted by TBA and (4) a variant of TBA (TBA-2strain) that implements genetic variation between BALB and C57 mouse strains, confirms TBA predictions. As a proof of principle, they show that (5) PPAR γ specifically affects Jun recruitment and ATF3 and JunD do not interact with PPAR γ in the absence of Jun.

The manuscript is a collection of large-scale data with little emphasis on communicating a converging and clear message. The novelty and impact of the results in terms of AP-1 in macrophage activation are questionable. The collaboration between AP-1 dimers and PPAR γ has been previously shown in the disease context (PMID:24411941; not cited by the authors). The cell-line specificity in terms of TF binding was shown by the ENCODE consortium (PMID: 22955990) so the JunD cistrome in different cell lines, although a good resource for testing TBA, does not bring any additional insights. Some major issues are:

1. The distinct regulatory roles of AP-1 family members cannot be concluded from a system where the differentially expressed genes in response to KLA are shown in TGEM whereas the hierarchical clustering and gene expression as a result of deleting AP-1 monomers are performed in iBMDMs. Immortalization is likely to affect AP-1 dimerization and the effect of thioglycolate on AP-1 expression and binding is not known nor shown. These controls must have been included in the study design and the experiments repeated in the same cell system, ideally primary macrophages. There are no technical details on the iBMDM generation in the methods. The results state that 'ATF3, Jun and JunD as the most expressed AP-1 family members under basal conditions' but Figure 1A shows JunD levels being similar to those of JunB in vehicle samples (same in supplementary figure 1a). In human macrophages JUN and FOSB are early LPS-response genes whereas mouse TGEMs do not up-regulate these significantly upon KLA treatment. How the results presented in Figure 2A and 5A can be interpreted to draw general conclusions on the AP-1 cistrome in macrophages?

2. Monomer binding sites: this definition is based on unique AP-1 family member Chip-Seq peaks and ChIP-seq was performed in basal and 1h KLA-treated TGEMs for ATF3, Jun, Fos, FosL2, Junb and JunD. As the authors state, there are 15 different AP-1 family members and the rationale behind performing ChIP-seq on these 6 AP-1 family members is based on their relative expression levels in

TGEMs (ATF3, Jun, JunD, JunB, FOSL2) and their KLA-responsiveness (FOS). The authors found that lowly expressed AP-1 members result in low number of binding sites but what about other highly expressed AP-1 members (ATF4) which are very likely to influence the global AP-1 binding landscape? ATF4 is the second most highly expressed ATF member in macrophages after ATF3 and it has been shown to be crucial for M-CSF signaling and osteoclast differentiation from mouse BMDMs (PMID:20628199), a process under AP-1 regulation.

3. If there are no differences in mRNA levels of AP-1 monomemers between BALB and C57BL6, how does TBE takes into account SNPs that may change their protein amounts in a post-transcriptional way ?

Minor:

1. Figure 5A is not cited correctly (page 11). The text refers to 178 mRNA increasing 2-fold but the figure shows the heatmap in the change of binding.

2. There is reference to a Figure 6H (page 15) which does not exist

3. The abstract is written in a conceptual and vague style. It is not clear what was not known before, what this paper investigated, which methods and cells were used and what was concluded and how the research complements the current understanding of AP-1 in macrophages.

4. There is no discussion on the potential biological impact of the findings. The collaboration between AP-1 and NF-KB is very well described in macrophage activation and Figure 5B confirms that. Even if the findings are confirmatory, the results should be discussed in the context of TLR4 activation in macrophages

Detailed responses to Reviewer comments:

Reviewer #1:

In this study, Fonseca et al. investigated the molecular bases for shared as well as specific ChIP-seq profiles of different AP-1 family members in basal and KLA-activated macrophages. Understanding how transcription factors with identical DNA binding specificities are recruited to different sites is in fact a classical and still unsolved problem that the authors tackled in a lucid and effective manner by combining machine learning approaches with mouse genetics.

The main conclusion of the study is that DNA binding specificities *in vivo* are the result of the availability of partner transcription factors that selectively promote recruitment of a specific AP-1 family member.

Overall, the approach, the data and the interpretation of the results are convincing and the manuscript represents an important contribution to the field.

Our response: We thank the reviewer for the positive review and recognition that we are working on a classical problem with implications for others working in the broad field of transcriptional regulation.

I have only a couple of comments for improvement:

1. It is unclear if the KO iBMDMs used represent different clones or polyclonal populations of deleted cells. Moreover, no details are provided regarding the generation of these immortalized cells.

Our response: The cells are a polyclonal population produced separately in duplicate using two distinct sets of sgRNA guides. We added the following text into the methods section: “KO iBMDMs were produced by infection of iBMDMs expressing CAS9 (iBMDM-CAS9-IRES-EGFP) with lenti viruses directing expression of guide RNAs specific for the AP-1 family members of interest. iBMDM-CAS9-IRES-EGFP cells were infected with MOI 100, as measured in 293T cells, with Lentiblast (5uL each reagent) in OPTI-MEM. Cells were then centrifuged at 1300g for 1h at room temperature. Media was then removed and cells were cultured in bone marrow media (30% L-cell, 20% FBS, 1% penicillin/streptomycin in DMEM) for two days. Infected cells were then purified by FACs sorting for expression of a transgene on the viral sequence (tagBFP2).”

2. One point I do not fully understand is why no motifs for the cooperating transcription factors identified by machine learning were not retrieved in a standard over-representation analysis (page 6, bottom). I assume this is due to the low fraction of specific events that are explained by each cooperating TF, but it would be important to clarify this issue.

Our response: We agree with the reviewer that it is important to clearly describe why our machine learning approach, TBA, can identify motifs that cannot be retrieved via standard over-representation analysis. The reviewer’s intuition is correct that motifs that occur at a low fraction of specific binding events may be difficult to pick up using over-representation analysis. There are two aspects of our approach that allow for better detection of motifs of cooperating TFs. First, our model can identify combinations of motifs that are co-enriched that cannot be identified using over-representation analysis, which compares the frequency of a motif at peaks versus background sequence individually. Consider a pair of motifs X and Y that individually occur at similar frequency in peaks and background sequences, which would not be identified using overrepresentation analysis. Supposing that motif X and Y co-occur frequently in just peaks, our model could assign X and Y moderate weights such that the occurrence of X and Y alone would not cause TBA to predict TF binding at a peak, and only the joint occurrence of X and Y would cause TBA to predict TF binding. Second, our model considers degenerate instances of motifs, allowing TBA to see more instances of a rare motif, whereas traditional approaches typically consider high affinity matches to a motif. We modified the

language used to describe our model to clarify this issue to readers who may have the same point of inquiry as the reviewer.

Reviewer #2:

Fonseca et al study the sequence features that are responsible for distinct DNA binding activities of different AP-1 family members in thioglycolate-elicited macrophages before and after stimulation with a TLR4 agonist. They do this by first identifying the binding sites of six AP-1 family members (ATF3, Jun, JunD, Fos, FosL2 and JunB) using ChIP-seq. They find that there are substantial differences in the binding patterns of all of the family members. To understand the reasons for these differences, they develop a simple logistic regression model to identify transcription factor motifs that are predictive for the binding of different AP-1 members. They also perform a number of validation experiments in immortalized bone-marrow derived macrophages confirm their findings. Overall, the paper is well written and provides novel insights into the binding specificities of AP-1 family members as well as confirming well-established models of collaborative interactions between TFs. The paper also highlights that a potentially large number of binding partners can influence the binding of a single TF - an observations that is currently not well appreciated in the field. However, all of these results rely on a crucial assumption that the antibodies used for the ChIP-seq experiments are highly specific to their target proteins. Moreover, important experimental details to judge the validity of this assumption are currently missing from the manuscript.

Our response: We appreciate the positive comments from the reviewer as well as the constructive criticism. We hope that our responses to concerns raised improve the clarity of the manuscript and better enable others to reproduce and build upon our results.

Major concerns

The quality of ChIP-seq experiments crucially depend on the quality and specificity of the antibodies used (<https://www.nature.com/news/reproducibility-crisis-blame-it-on-the-antibodies-1.17586>). Non-specific antibodies can lead to false positive results and multiple Nature group journal are now requiring the authors to fill a reporting checklist describing which antibodies were used and how they were validated. The current version of the manuscript does not contain any information about the antibodies used (manufacture name, lot number, etc) and how their specificity for the different AP-1 family members was validated. Thus, it is not clear if the differences in ChIP-seq signal observed between the six AP-1 family members are due to intrinsic binding properties of the TFs or non-specific binding of the antibodies to other transcription factors. At a minimum, the authors should clearly state which antibodies were used for the study and how their binding specificities were validated (either by the manufacturer or in the lab of the authors). If no evidence for validation can be provided, this should clearly be stated in the manuscript, explicitly acknowledging how this might influence the interpretation of the results.

Our response: This information was inadvertently deleted from the initial uploaded version of the manuscript. We now included details for all of the antibodies in Supplementary Table 3 (manufacturer and catalog number). Each antibody was specifically chosen to target regions of the protein sequence that were not conserved amongst the AP-1 family members that we targeted. We also added to the text to clarify this “Antibodies against AP-1 family members were chosen for targeting of non-conserved regions to minimize the potential for non-specific binding.

Similarly, the authors report that the PPAR half-site motif is specifically enriched among the ChIP-seq peaks of the Jun TF. However, the same result would also be consistent with a scenario in which

the Jun antibody exhibits low specificity for the PPAR γ factor which could drive the signal. The best validation would be to perform the ChIP-seq experiment for Jun in Jun-deficient cells to demonstrate that without the target protein present the ChIP-seq signal is reduced to background levels. However, if this is not feasible, the authors should seek other means of validation such as using a different antibody for Jun or demonstrating that in PPAR γ knockout cells the Jun signal decreases specifically at sites that are also bound by Jun and not at other PPAR γ binding sites across the genome.

Our response:

We thank Reviewer 2 for this suggestion. To provide evidence that the Jun ChIP-seq is specific to Jun and not PPAR γ , we performed ChIP-seq targeting Jun in iBMDM cells (in replicate) in which Jun has been knocked out using CRISPR. We were only able to detect 12 Jun peaks using HOMER (after filtering away peaks that had IDR < 0.05) in Jun knockout cells as compared to 25042 peaks in wildtype iBMDM cells treated with scramble control, demonstrating a high degree of specificity for the antibody. These data are now provided in a Supplement to figure 7.

Page 2, pg 3. The authors state that the experiments were performed in thioglycolate-elicited macrophages (TGEMs). However, it is not clear why TGEMs were chosen over other macrophages models. Moreover, it is not clear why the authors did not use BMDMs for their main experiments, given that this cell type was used for all of the knockout experiments. While I agree that this difference in cell types is unlikely to change the results, it would nevertheless be nice to have a justification for the experimental design somewhere in the paper. Secondly, since Nature Communications is a journal of general interest, it would be useful to add 1-2 sentence description of what the TGEMs are and how they are derived. People outside of the mouse macrophage field (but still interested in determinants of TF binding) might not be familiar with all of the different mouse macrophage models available.

Our response: BMDMs and TGEMs are both extensively used as model systems to study primary macrophages. Although the transcriptomes and epigenetic landscapes are not identical, we have shown that they are highly similar (Gosselin et al, 2014). Our experience has been that conclusions obtained from studies of TGEMs are broadly applicable to BMDMs and vice versa. We agree that it would be most consistent to have performed all of the studies in a single model system. However, after collection the majority of the ChIP-Seq data for each AP-1 factor in TGEMs, it became evident that it was not possible to achieve sufficiently robust knockdowns of these factors using conventional siRNA approaches. TGEMs were also resistant to lentiviral transduction required for CRISPR/CAS9 mediated mutagenesis. In order to perform loss of function studies to establish specific roles of AP-1 factors in macrophages it was necessary to employ iBMDMs as a model system that is amenable to CRISPR-Cas9 knockout methods. Importantly, these experiments are internally controlled by transducing Cas9 expressing iBMDMs with scrambled guide RNAs.

To make the manuscript more accessible, we added language to the results describing TGEMs. “To investigate the genome wide locations of AP-1 family members in a primary macrophage population, we generated thioglycollate elicited macrophages (TGEMs) by injection of thioglycollate into the peritoneal cavities of mice. Thioglycollate injection induces an inflammatory response, in which recruited macrophages become the major cell type by three days following injection. A highly purified population of TGEMs is obtained at this time by flushing the peritoneal cavity and allowing the macrophage population to adhere to a tissue culture plate.”

Minor issues

Throughout the main text of the paper, it is not clear which machine learning algorithm is being used. It is only in the discussion where the authors mention that they have used logistic regression. Although it is reasonable to leave implementation details for the Methods section, It would be beneficial to introduce the main learning algorithm earlier in the paper. For example, the caption for Figure 3 could contain this information.

Our response: We thank the reviewer for suggestions on how to improve the clarity of our manuscript, and we have revised our description of our approach in the results section and caption for Figure 3 to indicate that we used logistic regression. “TBA uses logistic regression to learn to distinguish the binding sites of a TF from a set of GC-matched background loci.”

Can you reproduce the PPARg result from the gkm-SVM model? For example, by looking at the k-mers that have high predictive power for Jun binding? Can you extract k-mer weights from the gkm-SVM model?

Our response: In the original manuscript describing gkm-SVM, the authors detailed a procedure that used the top 1% of the most highly ranked k-mers to calculate up to three de novo PWMs. In principle, the ranked k-mers can be used to identify additional motifs, and we have modified our discussion of gkm-SVM to indicate that this is potentially possible. To determine whether the PPARg half site motif can be retrieved using the gkm-SVM weights calculated for Jun, we quantified how well each k-mer matched to the PPARg half-site motif using the motif score. Of the 11-mers (default k is 11 for gkm-SVM) that were identified by gkm-SVM as positively correlated with Jun binding, we observed 74 k-mers that matched the PPARg half-site with a motif score greater than 10 and the median rank of these k-mers was 2406 out of 2097152 total 11-mers (best rank was 38, worst was 15041). As all these k-mers fall within the top 1% of k-mers (20971 highly ranked k-mers), we can conclude that the PPARg half site is recoverable using gkm-SVM. However, given the variability in how k-mers that matched the PPARg half-site were ranked, it is difficult for us to state how important the PPARg half-site is in comparison to other motifs that can be recovered from the k-mers ranked in the top 1% by gkm-SVM. While it is likely possible to devise a method to rank motifs recovered from the k-mers (which we indicate in our revised discussion section), we do not believe doing so would be consistent with the rest of the manuscript. The difficulty in interpreting rankings assigned to millions of k-mers is one of the motivations for why we devised our method. We state: “Efforts to build more advanced methods to extract information from machine learning models will allow not only for interpretation of future models of greater complexity, but also better understanding of existing models (Shrikumar 2017). For example, the procedure used by Ghandi and Lee et al to retrieve motifs from gkm-SVM can likely be improved to retrieve additional PWMs (Ghandi 2014)”

p19 line 6 from the bottom: Missing ‘to’ in: 'To assess the extent multi-collinearity in the motif score features we used to train our models.'

Our response: We corrected this error.

p20, pg 2. Since PPMs are matrices and not vectors, It is not clear how the Pearson correlation between the two motif is defined. Please clarify. Also, why did you decide the use Pearson correlation as the similarity measure for motif clustering, given that multiple other alternatives are available?

Our response:

We modified the methods section to clearly indicate how the Pearson correlation between two PPMs are calculated. Briefly, we flatten a pair of PPMs to form a pair of vectors which we then use to calculate the Pearson correlation. We initially considered each of the comparison metrics proposed by Mahony et al in their manuscript entitled “STAMP: a web tool for exploring DNA-binding motif similarities,” and ultimately selected Mahony et al’s default recommendation of the Pearson correlation for its intuitive behavior. The Pearson correlation similarity threshold we used to merge motifs together also showed a straightforward relationship to the maximum Variance Inflation Factor observed for the motif scores for the set of merged motifs, allowing us to easily identify a similarity threshold for merging together motifs that sufficiently reduces multiple collinearity. This Mahony et al reference was mistakenly omitted and is now included in our revised manuscript.

p20, pg 4. Please provide the exact versions of the ‘latest’ gkm-SVM, LS-GKM and BaMM software packages that were used in this study.

Our response: We modified both the main text as well as the methods to indicate the latest versions of software packages used. LS-GKM was compiled from source code downloaded from github.com/Dongwon-Lee/lsgkm on 08/25/16 and we used v1.0 of BaMM downloaded from github.com/soedinglab/BaMMmotif.

p17, pg 2 from bottom. In the discussion of more complex deep neural network models, you state that they are often difficult to interpret. While this is true in general, considerable progress has been made in this area recently. For example, you might find this recent paper from Peyton Greenside et al relevant (<https://www.biorxiv.org/content/early/2018/04/17/302711>).

Our response: We thank the reviewer for highlighting recent advances in extracting useful information from neural networks such as feature interactions. We modified the language in the discussion to indicate that progress in this direction will make deep neural network technology more accessible to genomics researchers. We too believe that neural networks can be interpreted, and we are engaged in separate efforts to build interpretable neural network models.

Reviewer #3:

Fonseca et al describe genome-wide transcriptional regulation of AP-1 in thioglycolate-elicited macrophages and immortalized bone marrow derived macrophages. They show that (1) AP-1 family members have distinct regulatory roles and can target monomer-specific loci in addition to overlapping ones (2) monomer-specific binding is not explained by differences in the DNA-binding domain but rather a differential interaction of locally-bound factors described as ensembles of collaborating TFs (PU.1, CEBP, RUNX). They then developed a machine learning approach (TBA – transcription factor Binding Analysis) with reduced multiple collinearity, which shows relatively higher performance when compared with existing sequence-based approaches with the exception of gkm-SVM. Outside the logical flow of the manuscript, the authors went on demonstrating the cell-type specific binding preferences of JunD. Finally, the manuscript shows that (3) TLR4 activation with KLA treatment reshapes the AP-1 cistrome which could be predicted by TBA and (4) a variant of TBA (TBA-2strain) that implements genetic variation between BALB and C57 mouse strains, confirms TBA predictions. As a proof of principle, they show that (5) PPAR γ specifically affects Jun recruitment and ATF3 and JunD do not interact with PPAR γ in the absence of Jun.

The manuscript is a collection of large-scale data with little emphasis on communicating a converging and clear message. The novelty and impact of the results in terms of AP-1 in macrophage

activation are questionable. The collaboration between AP-1 dimers and PPAR γ has been previously shown in the disease context (PMID:24411941; not cited by the authors). The cell-line specificity in terms of TF binding was shown by the ENCODE consortium (PMID: 22955990) so the JunD cistrome in different cell lines, although a good resource for testing TBA, does not bring any additional insights. Some major issues are:

Our response: The major objective of these studies was to investigate mechanisms by which members of a conserved family of transcription factors that bind to a common DNA recognition element are able to occupy both overlapping and distinct genomic locations and exert both redundant and non-redundant functions. The convergent and clear message that we hoped to convey was stated in the abstract as follows: 'These findings provide evidence that non-redundant genomic locations of different AP-1 family members in macrophages largely result from collaborative interactions with diverse, locus-specific ensembles of transcription factors and suggest a general mechanism for encoding functional specificities of their common recognition motif.' We are open to suggestions for how these points can be further clarified.

We thank Reviewer 3 for bringing the paper by Hasenfuss et al to our attention. This is an interesting study, but to our reading it does not provide any evidence for collaborative DNA binding interactions between AP-1 dimers and the PPAR γ transcription factor. Hasenfuss et al demonstrate that PPAR γ gene expression is regulated by members of the AP-1 family in the liver. Six different AP-1 family members are shown to bind to a common region of the PPAR γ promoter by locus specific ChIP. Gain and loss of function experiments indicate that some AP-1 heterodimers induce PPAR γ gene expression, while others are repressive. These differential functions have important biological consequences in a disease model of steatosis. However, there are no experiments in this paper that investigate the role of the PPAR γ transcription factor in differentially regulating the DNA binding properties of AP-1 family members. To our knowledge, there are no prior studies that systematically investigate mechanisms determining the distinct genomic binding profiles of specific AP-1 family members or members of other similarly conserved transcription factor families. We hope that with this clarification Reviewer 3 will agree that the current studies are both novel and significant.

We agree that the ENCODE consortium has demonstrated in a previous manuscript (PMID: 22955990) that a single TF can interact with different genomic regions in a cell type specific manner. As Reviewer 3 suggests, these data sets are primarily used as a way of additionally validating and extending the utility of the TBA algorithm.

1. The distinct regulatory roles of AP-1 family members cannot be concluded from a system where the differentially expressed genes in response to KLA are shown in TGEM whereas the hierarchical clustering and gene expression as a result of deleting AP-1 monomers are performed in iBMDMs. Immortalization is likely to affect AP-1 dimerization and the effect of thioglycolate on AP-1 expression and binding is not known nor shown. These controls must have been included in the study design and the experiments repeated in the same cell system, ideally primary macrophages.

Our response: As Reviewer 3 suggests, we have collected extensive transcriptomic and epigenetic data for TGEMs, BMDMs and iBMDMs under resting and activated conditions. While there are differences among each model system, BMDMs and TGEMs are highly similar to each other and distinct from tissue resident macrophages. Consistent with this, our experience has been that conclusions obtained from studies of TGEMs are broadly applicable to BMDMs and vice versa. We agree that it would be most consistent to have performed all of the studies in a single model system.

However, as noted in our response to Reviewer 1, after collection the majority of the ChIP-Seq data for each AP-1 factor in TGEMs, it became evident that it was not possible to achieve sufficiently robust knockdowns of these factors using conventional siRNA approaches. TGEMs were also resistant to lentiviral transduction required for CRISPR/CAS9 mediated mutagenesis. In order to perform loss of function studies to investigate specific roles of AP-1 factors in macrophages it was necessary to employ iBMDMs as a model system that is amenable to CRISPR-Cas9 knockout methods. Importantly, these experiments are internally controlled by transducing Cas9 expressing iBMDMs with scrambled guide RNAs. Therefore, factor-specific roles are clearly established in this model system, consistent with the prior findings of Hasenfuss in the liver. We clarify the limitation of extending these findings to TGEMs in the revised manuscript.

There are no technical details on the iBMDM generation in the methods.

Our response: We revised the methods text to detail how iBMDM cells were generated and added the corresponding reference (PMID: 2185941).

The results state that ‘ATF3, Jun and JunD as the most expressed AP-1 family members under basal conditions’ but Figure 1A shows JunD levels being similar to those of JunB in vehicle samples (same in supplementary figure 1a).

Our response: As noted by Reviewer 3, JunB mRNA levels are high. However, we were unable to detect JunB binding in 3 separate ChIP-seq experiments. In addition, we were unable to detect JunB by western blot in Vehicle treated conditions in the nucleus. We added a figure showing this western blot as well as the text “Despite high RNA expression in Veh treatment, JunB protein expression was not detected in the nucleus by western blot, explaining a lack of ChIP-seq signal (Supplementary Fig. 2B).”.

In human macrophages JUN and FOSB are early LPS-response genes whereas mouse TGEMs do not up-regulate these significantly upon KLA treatment. How the results presented in Figure 2A and 5A can be interpreted to draw general conclusions on the AP-1 cistrome in macrophages?

Our response: Transcription factor cistromes are established in a cell-specific manner based on genomic sequence and the expression levels and activities of collaborative transcription factors. We recently demonstrated, for example, that the cJun cistrome shows striking differences in macrophages derived from C57BL6, BALB, NOD, PWK and SPRET mice due to the influence of cis non coding genetic variation (Link et al, 2018). Differences in cis regulatory architecture appear to be major drivers of differences in human and mouse microglia gene expression (Gosselin et al, 2017). Conversely, the same macrophage lineage-determining factor, PU.1, binds to overlapping but distinct genomic locations in different tissue resident macrophage populations in the same strain of mice (Gosselin et al 2014). Therefore, each cistrome is highly context dependent. The general conclusion that we wish to present in this paper is that the overlapping and distinct cistromes of different AP-1 family members in a particular context are established by their differential interactions with combinations of co-expressed collaborative binding partners. The LPS-response in this case was primarily used as a strong perturbation of the expression and activities of these proteins that allows further testing of the factor-specific collaborative binding concept. We suggest that similar rules apply to AP-1 members in human macrophages, but due to differences in cis regulatory architecture and expression of collaborative transcription factors, specific cistromes will differ. We clarify these points in the revised manuscript.

2. Monomer binding sites: this definition is based on unique AP-1 family member Chip-Seq peaks and ChIP-seq was performed in basal and 1h KLA-treated TGEMs for ATF3, Jun, Fos, FosL2, Junb and JunD. As the authors state, there are 15 different AP-1 family members and the rationale behind performing ChIP-seq on these 6 AP-1 family members is based on their relative expression levels in TGEMs (ATF3, Jun, JunD, JunB, FOSL2) and their KLA-responsiveness (FOS). The authors found that lowly expressed AP-1 members result in low number of binding sites but what about other highly expressed AP-1 members (ATF4) which are very likely to influence the global AP-1 binding landscape? ATF4 is the second most highly expressed ATF member in macrophages after ATF3 and it has been shown to be crucial for M-CSF signaling and osteoclast differentiation from mouse BMDMs , a process under AP-1 regulation.

Our response: We modified Supplemental Figure 1A to include the expression level of ATF4, which is indeed highly expressed in comparison to the other AP-1 family members. This factor was omitted from the original figure because we were unable to obtain high quality ChIP-seq data for ATF4. We attempted ChIP-seq experiments targeting ATF4 after Vehicle and one hour KLA treatment using several antibodies and experimental protocols totaling to 6 attempts to perform ChIP-seq for ATF4 in each treatment condition. We added the following statement to the text: “Though ATF4 is highly expressed at the RNA level, we were unable to detect ATF4 by ChIP-seq using several conditions and several different antibodies”

3. If there are no differences in mRNA levels of AP-1 monomers between BALB and C57BL6, how does TBE takes into account SNPs that may change their protein amounts in a post-transcriptional way?

Our response: There are no variants in the protein coding sequences the AP-1 family members evaluated between BALB and C57BL6 mice that would directly result in differences in post transcriptional/post translational changes. In addition, we tested and found similar levels of protein expression among AP-1 family members in both Veh and KLA using western blots. This has been added to Supplementary Figure 6. Further, a reduction in protein expression in one strain in comparison to the other would result in a global decrease in binding. We observe that the number of sites that gained binding was roughly the same as the number of sites that lost binding when comparing BALB/cJ and C57BL/6J (as exemplified by Atf3 shown in Fig. 6A).

Minor:

1. Figure 5A is not cited correctly (page 11). The text refers to 178 mRNA increasing 2-fold but the figure shows the heatmap in the change of binding.

Our response: We thank Reviewer 3 for pointing out this error, which is now corrected in the revised manuscript.

2. There is reference to a Figure 6H (page 15) which does not exist

Our response: We thank Reviewer 3 for pointing out this error, which is now corrected in the revised manuscript.

3. The abstract is written in a conceptual and vague style. It is not clear what was not known before, what this paper investigated, which methods and cells were used and what was concluded and how the research complements the current understanding of AP-1 in macrophages.

Our response: The conceptual style of the abstract is enforced by the 150 word limit stipulated by the journal.

We summarized what is not known in the first sentence:

Mechanisms by which members of the AP-1 family of transcription factors play both redundant and non-redundant biological roles despite recognizing the same DNA sequence remain poorly understood. (27 words)

We summarized what this paper investigated, the methods and main results by the following four sentences:

To address this question, we investigated the molecular functions and genome-wide DNA binding patterns of AP-1 family members in macrophages. ChIP-sequencing showed overlapping and distinct binding profiles for each factor that were remodeled following TLR4 ligation. Development of a machine learning approach that jointly weighs hundreds of DNA recognition elements yielded dozens of motifs predicted to drive factor-specific binding profiles. Machine learning-based predictions were confirmed by analysis of the effects of mutations in genetically diverse mice and by loss of function experiments. (82 words)

We summarized the major conclusions with the following sentence:

These findings provide evidence that non-redundant genomic locations of different AP-1 family members in macrophages largely result from collaborative interactions with diverse, locus-specific ensembles of transcription factors and suggest a general mechanism for encoding functional specificities of their common recognition motif. (41 words)

Total 150 words

We are open to suggestions for improving the abstract.

4. There is no discussion on the potential biological impact of the findings. The collaboration between AP-1 and NF- κ B is very well described in macrophage activation and Figure 5B confirms that. Even if the findings are confirmatory, the results should be discussed in the context of TLR4 activation in macrophages.

Our response: We discussed potential biological impact of the findings in the final paragraph of the discussion as follows:

Collectively, our findings suggest two classes of collaborative TFs: 1) highly ranked TFs that are strongly correlated with the binding of all AP-1 monomers, including TFs important to macrophage identity such as such as PU.1 and C/EBPs (Fig. 4A, black and grey boxes), and 2) moderately ranked TFs that specify the binding of individual AP-1 monomers (Fig. 4D, red and blue boxes). The former likely consists of TFs that play a role in opening chromatin while the latter class of TFs may allow for tuning the optimal level of transcriptional activation or response. These two classes of motifs were also seen in TLR4 activated macrophages where highly ranked motifs, such as Nf κ B, were correlated with the binding of all AP-1 family members (Supp Table 1), while a large set of moderately ranked motifs distinguished each AP-1 monomer (Supplementary Fig. 5C). Overall, these studies provide

evidence that collaborative interactions of TFs allow a single DNA motif to be used in a wide variety of contexts, which may be a general principle for how transcriptional specificity is encoded by the genome.

We are open to suggestions for improving this section.

Reviewer #1 (Remarks to the Author):

The authors have fully addressed my issues and now I strongly recommend this manuscript for publication

Reviewer #2 (Remarks to the Author):

The authors have fully address all of my concerns.

I have only one more minor comment: The caption for Supplementary Figure 7 currently states 250,041 peaks detected in scramble IBMDMs while the response to reviewers states 25,042. I assume this is a typo?

Reviewer #3 (Remarks to the Author):

In their revised manuscript, the authors clarified the following points:

1. Transcription factor cistrome dependency on cell type, genomic sequence and the expression levels and activities of collaborative transcription factors. The context-dependency of the cistromes and the significance of the results in murine cells when compared to human macrophages were clarified in the revised text.

2. Despite high mRNA levels, an undetectable JunB protein in vehicle samples by Western Blot explaining the lack of Chip-seq. The authors clarified this by adding a Western Blot in Supplementary Figure 2B.

3. The iBMDM generation methods and a discussion on the biological impact of the findings were included. Furthermore, supplementary Figure 6 shows no protein differences between the two strain of mice, which argues against any post-transcriptional changes.

However the issue of distinct model systems (TGEM vs iBMDMs, Figure 1) remains and is likely to influence the interpretation of the results. The lack of ChIP-seq signal for ATF4 (the third most expressed AP-1 member, Supplementary Figure 1a) precludes from drawing a conclusion on global AP-1 binding landscape. The abstract was not modified to include the cell type (TGEM, iBMDM).

Detailed responses to Reviewer comments:

REVIEWERS' COMMENTS:

Reviewer #1 (Remarks to the Author):

The authors have fully addressed my issues and now I strongly recommend this manuscript for publication

Our response: We thank the reviewer for the thoughtful reviews.

Reviewer #2 (Remarks to the Author):

The authors have fully address all of my concerns.

I have only one more minor comment: The caption for Supplementary Figure 7 currently states 250,041 peaks detected in scramble iBMDMs while the response to reviewers states 25,042. I assume this is a typo?

Our response: We thank the reviewer for the thoughtful reviews. We have fixed the caption in Supplementary Figure 7 to read 25,041 peaks.

Reviewer #3 (Remarks to the Author):

In their revised manuscript, the authors clarified the following points:

1. Transcription factor cistrome dependency on cell type, genomic sequence and the expression levels and activities of collaborative transcription factors. The context-dependency of the cistromes and the significance of the results in murine cells when compared to human macrophages were clarified in the revised text.
2. Despite high mRNA levels, an undetectable JunB protein in vehicle samples by Western Blot explaining the lack of Chip-seq. The authors clarified this by adding a Western Blot in Supplementary Figure 2B.
3. The iBMDM generation methods and a discussion on the biological impact of the findings were included. Furthermore, supplementary Figure 6 shows no protein differences between the two strain of mice, which argues against any post-transcriptional changes.

However the issue of distinct model systems (TGEM vs iBMDMs, Figure 1) remains and is likely to influence the interpretation of the results. The lack of ChiP-seq signal for ATF4 (the third most expressed AP-1 member, Supplementary Figure 1a) precludes from drawing a conclusion on global AP-1 binding landscape. The abstract was not modified to include the cell type (TGEM, iBMDM).

Our response: We thank the reviewer for the thoughtful reviews. We agree that iBMDMs and TGEMs are different environments. However, we do believe that the AP-1 knockout data in iBMDMs in Figure 1 conclusively shows that AP-1 family members have distinct effects on the RNA transcriptome and that this result translates to other contexts as shown for JunD in Supplementary Figure 4 C,D. Similarly, though a high quality ATF4 ChIP-seq would have allowed us to draw clearer conclusions on the global AP-1 binding landscape, we believe that our results are representative of

the binding of the AP-1 family in TGEMs . Specifically, we are still able to differentiate family member binding and conclude differences are a result of cooperative binding with other transcription factors. Both of these issues are unfortunate technical limitations.

We have added " genome-wide DNA binding patterns of AP-1 family members in primary and immortalized mouse macrophages" to include reference to both TGEMs and iBMDMs in the available space.